# Rethinking Incentives in Recommender Systems: Are Monotone Rewards Always Beneficial?

**Fan Yao**
University of Virginia
fy4bc@virginia.edu

**Chuanhao Li**
Yale University
chuanhao.li.cl2637@yale.edu

**Karthik Abinav Sankararaman**
Meta
karthikabinavs@meta.com

**Yiming Liao**
Meta
yimingliao@meta.com

**Yan Zhu**
Google
yanzhuyz@google.com

**Qifan Wang**
Meta
wqfcr@meta.com

**Hongning Wang**
University of Virginia
hw5x@virginia.edu

**Haifeng Xu**
University of Chicago
haifengxu@uchicago.edu

## Abstract

The past decade has witnessed the flourishing of a new profession as media content creators, who rely on revenue streams from online content recommendation platforms. The reward mechanism employed by these platforms creates a competitive environment among creators which affects their production choices and, consequently, content distribution and system welfare. It is thus crucial to design the platform's reward mechanism in order to steer the creators' competition towards a desirable welfare outcome in the long run. This work makes two major contributions in this regard: first, we uncover a fundamental limit about a class of widely adopted mechanisms, coined *Merit-based Monotone Mechanisms*, by showing that they inevitably lead to a constant fraction loss of the optimal welfare. To circumvent this limitation, we introduce *Backward Rewarding Mechanisms* (BRMs) and show that the competition game resultant from BRMs possesses a potential game structure. BRMs thus naturally induce strategic creators' collective behaviors towards optimizing the potential function, which can be designed to match any given welfare metric. In addition, the class of BRM can be parameterized so that it allows the platform to directly optimize welfare within the feasible mechanism space even when the welfare metric is not explicitly defined.

## 1 Introduction

Online recommendation platforms, such as Instagram and YouTube, have become an integral part of our daily life [5]. Their impact extends beyond merely aligning users with the most relevant content: they are also accountable for the online ecosystem it creates and the long-term welfare it promotes, considering the complex dynamics driven by the potential strategic behaviors of content creators [17]. Typically, creators' utilities are directly tied to the visibility of their content or economic incentives they can gather from the platform, and they constantly pursue to maximize these benefits [10, 12]. This fosters a competitive environment that may inadvertently undermine the *social welfare*, i.e., the total utilities of all users and content creators in the system [9]. For example, consider a scenario where the user population contains a majority group and many smaller minority groups, where different groups are interested in distinct topics. The social welfare is maximized when content distribution covers the variety of topics. However, a possible equilibrium of this competition can lead

37th Conference on Neural Information Processing Systems (NeurIPS 2023).

most content creators to produce homogeneous content catering only to the majority group. This is because the benefits from creating niche content cannot offset the utility loss caused by forgoing the exposure from the majority of users. Such a phenomenon could potentially dampen the engagement of minority user groups or even instigate them to leave the platform altogether. This can consequently hurt the overall social welfare and also impact the platform's long-term revenue.

To counter such effects induced by strategic content creators, the platform can design proper incentives (i.e., rewarding mechanisms) to influence the creators' perceived utilities, thereby steering the equilibrium content distribution towards enhanced social welfare. In reality, many platforms share revenue with creators via various mechanisms [15, 19, 24, 20]. These incentives are typically proportional to user satisfaction measured by various metrics, such as click-through rate and engagement time. We model such competitive environment within a general framework termed *content creator competition* ($C^3$) game that generalizes and abstracts a few established models including [23, 4, 14, 13], and frame a class of popular rewarding mechanisms as Merit-based Monotone Mechanisms ($\mathcal{M}^3$). The $\mathcal{M}^3$ are characterized by a few simple properties, intuitively meaning better content should be rewarded more (i.e., merit-based) and sum of creators' utilities increase whenever any creator increases her content relevance (i.e., monotone). These properties reflect the essence of most employed rewarding mechanisms in practice. However, we show that $\mathcal{M}^3$ necessarily incur a constant fraction of welfare loss in natural scenarios due to failing to encourage content creators who are content with generating popular content for majority user groups to produce niche content.

This surprising negative result uncovers the intrinsic *incompatibility* within $\mathcal{M}^3$ mechanisms and thus compels us to rethink the incentive design in recommender systems (RS). A key property of $\mathcal{M}^3$ is monotonicity, which stipulates that when the matching quality of exposed creators to a specific user group exhibits a Pareto improvement, the total reward received by those creators also increases. We point out that, while seemingly plausible at the first thought, this property undesirably encourages excessive concentration of creators around the majority user groups and leaves minority groups underserved. To resolve this issue, we question the validity of this monotone property. At a high level, when creators' competition within some user group surpasses a limit that begins to harm welfare, the platform should reduce their total gain. In light of this insight, we introduce a new class of content rewarding mechanism coined the *Backward Rewarding Mechanisms* (BRM), which drops monotonicity but remains merit-based. The strength of BRM lies in three aspects: **1.** any $C^3$ game under any BRM mechanism forms a potential game [16]; **2.** we can identify a BRM mechanism such that the induced potential function is equivalent to any given social welfare metric; consequently, the net effect of creators' competition aligns perfectly with maximizing the social welfare; and **3.** BRM contains a parameterized subclass of mechanisms that allows empirical optimization of the social welfare, which is especially useful in practice when the welfare is not explicitly defined. These merits of BRM are supported by our empirical studies, in which we developed simulated environments, demonstrating the welfare induced by BRM outperforms baseline mechanisms in $\mathcal{M}^3$.

## 2   Related Work

The studies of content creators' strategic behavior under the mediation of an RS starts from the original work of [3, 4], who proposed the Shapley mediator that guarantees the existence of pure Nash equilibrium (PNE) and several fairness-related requirements. These works only study the design of the content-user matching probability, and it was observed that user welfare could be significantly compromised. In contrast, our work considers the design of another important "knob" of contemporary platforms — i.e., the reward for each content creator. We propose a broad class of rewarding mechanisms, namely, the Backward Rewarding Mechanisms (BRM). We show that the Shapley mediator of [4] turns out to be an example of our BRM class; however, by optimizing within the class of BRM mechanisms, the RS can now achieve the goal of maximizing social welfare.

Several recent work [13, 14, 23] studied the properties of creator-side equilibrium in the $C^3$ game, under given creator incentives. In [13, 14], creators are assumed to directly compete for user *exposure* without the mediation of an RS. These studies focus on characterizing the Nash Equilibrium (NE) and identifying conditions that may trigger specialization among creators' strategies. [23] demonstrate that the user welfare loss under a conventional RS using top-$K$ ranking is upper-bounded by $O(\frac{1}{\log K})$ when creators compete for user *engagement*. Our work reveals that any merit-based monotone mechanism, including but not limiting to those based on user exposure or engagement, will inevitably incur at least a $\frac{1}{K}$ fraction of welfare loss. However, should the platform can design creators' incentive signals, then exactly optimal social welfare could be obtained.

The main goal of the present work is to design incentives for creators to steer their collective behaviors towards social optimum. Such welfare-maximizing mechanism design has been studied extensively in social choice theory as well as in recent algorithmic mechanism design literature. Two of the most fundamental findings in this space are perhaps: (1) Vickrey–Clarke–Groves (VCG) mechanism which maximizes the social welfare of multi-item allocation [22]; and (2) the Arrow's impossibility theorem which employs an axiomatic approach to show the impossibility of welfare-maximization among natural voting mechanisms [1].[1] While welfare maximization in resource allocation and social choice has been extensively studied, to the best of our knowledge, our work is the first study of designing optimal mechanisms for welfare maximization in *recommender systems*. Interestingly, both our negative and positive results are inspired by the two fundamental results mentioned above. [1] shows that there is no ranked voting system capable of transforming the ranked preferences of individuals into a communal consensus ranking, while maintaining a natural set of criteria. Drawing a parallel to this concept and using the same axiomatic approach, our Theorem 1 can be interpreted as a similar impossibility result for welfare maximization under certain axioms in recommender systems — that is, no rewarding mechanism is capable of optimizing the social welfare while adhering to both "merit-based" and "monotone" properties. On the other hand, our positive result shows that there exists a creator rewarding mechanism that can maximize the RS's welfare at the potential-function-maximizing pure Nash equilibrium. The conceptual message of this result bears similarity to VCG's welfare maximization in multi-item allocation at the dominant-strategy equilibrium, but the techniques we employed is significantly different from VCG. To the best of our knowledge, this appear the first attempt to employ potential functions for welfare-maximizing mechanism design.

## 3 A General Model for Content Creator Competition

In this section, we formalize the Content Creator Competition ($C^3$) game as well as the platform's rewarding mechanisms. Each $C^3$ game instance $\mathcal{G}$ can be described by a tuple $(\{\mathcal{S}_i\}_{i=1}^n, \{c_i\}_{i=1}^n, \mathcal{F}, \sigma, M, \{r_i\}_{i=1}^n)$ illustrated as follows:

1. **Basic setups:** The system has a user population/distribution $\mathcal{F}$ with (discrete or continuous) support $\mathcal{X} \subset \mathbb{R}^d$, and a set of content creators denoted by $[n] = \{1, \cdots, n\}$. Each creator $i$ can take an action $\boldsymbol{s}_i$, often referred to as a *pure strategy* in game-theoretic terms, from an action set $\mathcal{S}_i \subset \mathbb{R}^d$. Any $\boldsymbol{s}_i \in \mathcal{S}_i$ can be interpreted as the embedding of a content that creator $i$ is able to produce. Let $c_i(\boldsymbol{s}_i)$ denote the production cost for creator $i$ to generate $\boldsymbol{s}_i$. As an example, one may think of $c_i(\boldsymbol{s}_i) = \lambda_i \|\boldsymbol{s}_i - \bar{\boldsymbol{s}}_i\|_2^2$ where $\bar{\boldsymbol{s}}_i$ represents the type of content that creator $i$ is most comfortable or confident with, though our result is general and does not depend on any assumption of $c_i$. In general, any creator $i$ may also play a mixed strategy, i.e., a distribution over $\mathcal{S}_i$. However, for the purpose of this study, it suffices to consider pure strategies since it always exists in all our analysis [2] and thus is a more natural solution concept.

   The connection between any user $\boldsymbol{x}$ (drawn from $\mathcal{F}$) and content $\boldsymbol{s}_i$ is described by a matching score function $\sigma(\boldsymbol{s}; \boldsymbol{x}) : \mathbb{R}^d \times \mathbb{R}^d \to \mathbb{R}_{\geq 0}$ which measures the matching quality between a user $\boldsymbol{x} \in \mathcal{X}$ and content $\boldsymbol{s}$. Without loss of generality, we normalize $\sigma$ to $[0, 1]$, where 1 suggests perfect matching. This work focuses on modeling the strategic behavior of creators, thus abstracts away the estimation of $\sigma$ and simply views it as perfectly given.[3] With slight abuse of notation, we use $\sigma_i(\boldsymbol{x})$ to denote $\sigma(\boldsymbol{s}_i; \boldsymbol{x})$ given any joint creator action profile $\boldsymbol{s} = (\boldsymbol{s}_1, \cdots, \boldsymbol{s}_n) \in \mathcal{S} = \cup_{i=1}^n \mathcal{S}_i$. When it is clear from the context, we often omit the reference to the generic user $\boldsymbol{x}$ (drawn from population $\mathcal{F}$) and simply use $\sigma_i$ to denote creator $i$'s score.

2. **Rewarding mechanisms and resultant creator utilities.** Given joint strategy $\boldsymbol{s} = (\boldsymbol{s}_1, \cdots, \boldsymbol{s}_n) \in \mathcal{S}$, the platform generates a reward $u_i \in [0, 1]$ for each user-creator pair $(\boldsymbol{s}_i, \boldsymbol{x})$. We generally allow $u_i$ to depend on $\boldsymbol{s}_i$'s matching score $\sigma_i$ and also other creators' score $\sigma_{-i} = \{\sigma_t | 1 \leq t \leq n, t \neq i\}$. Thus, a rewarding mechanism $M$ is a mapping from $(\sigma_i, \{\sigma_{-i}\})$ to $[0, 1]$, which is denoted by the function $M(\sigma_i; \sigma_{-i})$. Such rewarding mechanisms can be interpreted as the expected payoff for creator $i$ under any user-content matching strategy *and* some post-matching rewarding scheme. For example, suppose the platform matches creator $\boldsymbol{s}_i$ to

---

[1]The term "welfare" in social choice is classically more concerned with fairness or stability, as opposed to utility maximization.

[2]We will propose mechanisms that induce a potential game structure, which guarantees the existence of PNE[18].

[3]It is an interesting open question of studying how our results can be extended to the situation in which the estimation of $\sigma$ is inaccurate or has bias, though this is out of the scope of the present paper.

user $\boldsymbol{x}$ with probability $p(\sigma_i; \sigma_{-i})$ and then reward each matched creator-$i$ by some $R_i$. Then by letting $R_i = \mathbb{I}[\boldsymbol{s}_i \text{ matched to } \boldsymbol{x}] \frac{M(\sigma_i; \sigma_{-i})}{p(\sigma_i; \sigma_{-i})}$ we have $\mathbb{E}[R_i] = M(\sigma_i; \sigma_{-i})$. Given such correspondence between expected creator payoff and user-content matching/post-matching reward, we can without loss of generality refrain from the modeling of detailed matching policy and rewarding schemes, and simply focus on the design of $M(\cdot; \cdot)$.

A few remarks about the reward mechanism $M(\cdot; \cdot)$ follow. First, $M$ is determined only by the profile of matching scores but not directly depend on the specific user $\boldsymbol{x}$. However, our main results can be seamlessly generalized to allow $M$ directly depend on $\boldsymbol{x}$.[4] Second, the definition of $M(\sigma; \sigma_-)$ above naturally implies that it is "identity-invariant". That is, it specifies the reward of a matching score $\sigma$, generated by whichever creator, when facing a set of competitive matching scores in $\sigma_-$. While one could have considered more general identity-dependent rewarding mechanisms, they appear less realistic due to fairness concerns. More importantly, we shall show that such identity-invariant mechanisms already suffice to achieve optimal welfare.

Under the rewarding mechanism above, creator-$i$'s expected utility is simply the expected reward gained from the user population $\mathcal{F}$ minus the cost for producing content $\boldsymbol{s}_i$, i.e.,

$$u_i(\boldsymbol{s}) = \mathbb{E}_{\boldsymbol{x} \in \mathcal{F}}[M(\sigma_i(\boldsymbol{x}); \sigma_{-i}(\boldsymbol{x}))] - c_i(\boldsymbol{s}_i), \forall i \in [n], \tag{1}$$

where $\sigma_i(\boldsymbol{x}) = \sigma(\boldsymbol{s}_i; \boldsymbol{x})$ is the matching score between $\boldsymbol{s}_i, \boldsymbol{x}$ and $c_i$ is the cost function for creator-$i$.

3. **User utility and the social welfare.** Before formalizing the welfare objective, we first define a generic user $\boldsymbol{x}$'s utility from consuming a list of ranked content. Since the user attention usually decreases in the rank positions, we introduce discounting weights $\{r_k \in [0, 1]\}_k$ to represent his/her "attention" over the $k$-th *ranked* content. Naturally, we assume $r_1 \geq \cdots \geq r_{n-1} \geq r_n$, i.e., higher ranked content receives more user attention. Consequently, the user's utility from consuming a list of content $\{l(k)\}_{k=1}^n$, which is a permutation of $[n]$ ranked in a descending order of match scores (i.e., $\sigma_{l(1)} \geq \sigma_{l_j(2)} \geq \cdots \geq \sigma_{l_j(n)}$), is defined by the following weighted sum

$$W(\boldsymbol{s}; \boldsymbol{x}) = \sum_{k=1}^n r_k \sigma_{l(k)}(\boldsymbol{x}). \tag{2}$$

We provide additional examples that account for top-$K$ ranking rules with arbitrary ad-hoc permutations in Appendix 8.1. Finally, the social welfare is defined as the sum of total user utilities and total creator utilities, minus the platform's cost:

$$W(\boldsymbol{s}; \{r_k\}) = \mathbb{E}_{\boldsymbol{x} \sim \mathcal{F}}[W(\boldsymbol{s}; \boldsymbol{x})] + \sum_{i=1}^n u_i(\boldsymbol{s}) - \sum_{i=1}^n \mathbb{E}_{\boldsymbol{x} \sim \mathcal{F}}[M(\sigma_i(\boldsymbol{x}); \sigma_{-i}(\boldsymbol{x}))]$$

$$= \mathbb{E}_{\boldsymbol{x} \sim \mathcal{F}}[W(\boldsymbol{s}; \boldsymbol{x})] - \sum_{i=1}^n c_i(\boldsymbol{s}_i). \tag{3}$$

The set of weights $\{r_k\}$ determines a welfare metric $W(\cdot; \{r_k\})$. For ease of exposition, we assume the sequence $\{r_k\}$ is independent of specific user $\boldsymbol{x}$. However, our results also hold for the more general situation where $\{r_k\}$ is a function of the user profile $\boldsymbol{x}$. In most of our analysis, we assume $r_k$ can be measured and is known to the platform. However, we will later discuss how to address the situations where the platform only has blackbox access to $W(\cdot, \{r_k\})$, but not the individual values of $r_k$.

**The research question: creator incentive design for welfare maximization.** Unlike previous works [2, 13, 14] that primarily focus on designing user-content matching mechanisms, we consider the design of a different, and arguably more general, "knob" to improve the system's welfare, i.e., creators' rewarding schemes. Each rewarding mechanism $M$ establishes a competitive environment among content creators, encapsulated by a $C^3$ instance $\mathcal{G}(\{\mathcal{S}_i\}, \{c_i\}, \mathcal{F}, \sigma, M, \{r_i\})$. To characterize the outcome of $C^3$ game, we consider the solution concept called Pure Nash Equilibrium (PNE), which is a joint strategy profile $\boldsymbol{s}^* = (\boldsymbol{s}_1^*, \cdots, \boldsymbol{s}_n^*) \in \mathcal{S}$ such that each player $i$ cannot increase his/her utility by unilaterally deviating from $\boldsymbol{s}_i^*$. Our objective is thus to design mechanisms $M$ that: 1. guarantees the existence of PNE, thereby ensuring a stable outcome, and 2. maximizes social welfare at the PNE. In the upcoming sections, we first demonstrate why many existing rewarding mechanisms can fall short of achieving these goals, and then introduce our proposed new mechanism.

---

[4]This may be useful when the system wants to specifically promote a particular user group by providing higher rewards to creators for serving this group.

# 4  The Fundamental Limit of Merit-based Monotone Mechanisms

In this section, we employ an axiomatic approach to demonstrate the fundamental limit of many employed rewarding mechanisms in today's practice. We identify a few properties (sometimes also called *axioms* [1]) of rewarding mechanisms that are considered natural in many of today's RS platforms, and then show that any mechanism satisfying these properties will necessarily suffer at least $1/K$ fraction of welfare loss at every equilibrium of some natural RS environments. Specifically, we consider mechanisms with the following properties.

**Definition 1** (Merit-based Monotone Mechanisms ($\mathcal{M}^3$)). *We say $M$ is a* merit-based monotone mechanism *if for any relevance scores $1 \geq \sigma_1 \geq \cdots \geq \sigma_n \geq 0$, $M$ satisfies the following properties:*

- *Merit-based:*

    - *(Normality) $M(0; \sigma_{-i}) = 0$, $M(1; \{0, \cdots, 0\}) > 0$,*
    - *(Fairness) $M(\sigma_i; \sigma_{-i}) \geq M(\sigma_j; \sigma_{-j}), \forall i > j$,*
    - *(Negative Externality) $\forall i$, if $\sigma_{-i} \preccurlyeq \sigma'_{-i}$ ($\sigma_j \leq \sigma'_j, \forall j \neq i$), then $M(\sigma_i; \sigma_{-i}) \geq M(\sigma_i; \sigma'_{-i})$.*

- *Monotonicity: the total rewards $\sum_{i=1}^n M(\sigma_i; \sigma_{-i}) : [0,1]^n \to \mathbb{R}_{\geq 0}$ is non-decreasing in $\sigma_i$ for every $i \in [n]$.*

*We use $\mathcal{M}^3(n)$ to denote the set of all merit-based monotone mechanisms. When the context is clear, we omit the argument $n$ and simply use the notation $\mathcal{M}^3$.*

The two properties underpinning $\mathcal{M}^3$ are quite intuitive. Firstly, the merit-based property consists of three natural sub-properties: 1. zero relevance content should receive zero reward, whereas the highest relevance content deserves a non-zero reward; 2. within the given pool of content with scores $\{\sigma_i\}_{i \in [n]}$, the higher relevance content should receive a higher reward; 3. any individual content's reward does not increase when other creators improve their content relevance. Secondly, monotonicity means if any content creator $i$ improves her relevance $\sigma_i$, the total rewards to all creators increase. This property is naturally satisfied by many widely adopted rewarding mechanisms because platforms in today's industry typically reward creators proportionally to user engagement or satisfaction, the *total* of which is expected to increase as some creator's content becomes more relevant.

Indeed, many popular rewarding mechanisms can be shown to fall into the class of $\mathcal{M}^3$. For instances, the following two mechanisms defined over a descending score sequence $\{\sigma_i\}$ are widely adopted in current industry practices for rewarding creators [15, 19, 20, 24], both of which are in $\mathcal{M}^3$:

1. When players' utilities are set to the total content exposure [2, 13, 14], we have $M(\sigma_i; \sigma_{-i}) = \mathbb{I}[i \leq K]\frac{\exp(\beta^{-1}\sigma_i)}{\sum_{j=1}^K \exp(\beta^{-1}\sigma_j)}$, with a temperature parameter $\beta > 0$ controlling the spread of rewards.

2. When players' utilities are set to the user engagement [23], we have $M(\sigma_i; \sigma_{-i}) = \mathbb{I}[i \leq K]\frac{\exp(\beta^{-1}\sigma_i)}{\sum_{j=1}^K \exp(\beta^{-1}\sigma_j)}\pi(\sigma_1, \cdots, \sigma_n)$, where $\pi(\sigma_1, \cdots, \sigma_n) = \beta \log\left(\sum_{j=1}^K \exp(\beta^{-1}\sigma_j)\right)$ is shown to be the total user welfare.

We show that *any* mechanism in $\mathcal{M}^3$ may result in quite suboptimal welfare, even applied to some natural $C^3$ game environment. We consider the following representative (though idealized) sub-class of $C^3$ instances, which we coin the *Trend v.s. Niche* (TvN) environments. As outlined in the introduction section, TvN captures the essence of many real-world situations.

**Definition 2** (TvN Games). *The* Trend v.s. Niche *(TvN) game is specified by the following RS environments:*

- *The user population $\mathcal{F}$ is a uniform distribution on $\mathcal{X} = \{\boldsymbol{x}_j\}_{j=1}^{2n}$ where $\boldsymbol{x}_j = \boldsymbol{e}_1$, for $1 \leq j \leq n+1, \boldsymbol{x}_{n+2} = \boldsymbol{e}_2, \cdots, \boldsymbol{x}_{2n} = \boldsymbol{e}_n$ and $E = \{\boldsymbol{e}_1, \cdots, \boldsymbol{e}_n\} \subset \mathbb{R}^n$ is the set of unit basis in $\mathbb{R}^n$;*

- *All creators have zero costs and share the same action set $\mathcal{S}_i = E$; the relevance is measured by the inner product, i.e., $\sigma(\boldsymbol{s}; \boldsymbol{x}) = \boldsymbol{s}^\top \boldsymbol{x}$;*

- *The attention discounting weights $\{r_i\}$ is induced by a top-$K$ environment, i.e., $r_1 \geq \cdots \geq r_K \geq r_{K+1} = \cdots = r_n = 0$.*

*The content creation competition game induced by any mechanism $M$ is called a TvN game, denoted as $\mathcal{G}(\{\mathcal{S}_i\}, \{c_i = 0\}, \mathcal{F}, \sigma, M, \{r_i\})$.*

The TvN game models a scenario where the user population comprises multiple interest groups, each with orthogonal preference representations. In this game, the largest group consists of nearly half the population. Each content creator has the option to cater to one—and only one—user group. While this game is simple and stylized, it captures the essence of real-world user populations and the dilemmas faced by creators. Creators often find themselves at a crossroad: they must decide whether to pursue popular trends for a broader audience population, leading to intense competition, or focus on niche topics with a smaller audience and reduced competition. Our subsequent result shows that if the platform adopts $\mathcal{M}^3$ in the TvN game, this tension of content creation turns out to be a curse in the sense that a unique PNE is achieved when all players opt for the same strategy — catering to the largest user group — and we quantify the social welfare loss at this PNE in the following.

**Theorem 1.** *For any rewarding mechanism $M \in \mathcal{M}^3$ applied to any TvN instance, we have*

1. *the resultant game admits a unique NE $\boldsymbol{s}^*$;*

2. *the welfare of this NE is at most $\frac{K}{K+1}$ fraction of the optimal welfare for large $n$. Formally,*

$$\frac{W(\boldsymbol{s}^*)}{\max_{\boldsymbol{s} \in \mathcal{S}} W(\boldsymbol{s})} \le \frac{K}{K+1} + O\left(\frac{1}{n}\right). \tag{4}$$

The proof is in Appendix 8.4, where we explicitly characterize both $\boldsymbol{s}^*$ and the welfare maximizing strategy profile and calculate their difference in terms of welfare. It is worthwhile to point out that the reciprocal of left-hand side of (4) is commonly known as the Price of Anarchy (PoA). This metric gauges the welfare loss at equilibrium compared to optimal welfare. (4) suggests that the PoA of $\mathcal{G}$ under $\mathcal{M}^3$ could be as significant as $1/2$ for users who primarily care about the top relevant content, which is shown to be realistic given the diminishing attention spans of Internet users [6]. This theorem shows that no mechanisms in $\mathcal{M}^3$ can achieve the optimal welfare at the (unique) equilibrium of any TvN game. This naturally motivates our next question about how to design welfare-maximizing rewarding mechanisms in recommender systems.

## 5 Welfare Maximization via Backward Rewarding Mechanisms

The aforementioned failure of the generic $\mathcal{M}^3$ class for welfare maximization urges us to re-think the rewarding mechanism design in recommender systems, especially for platforms where user attention is concentrated on the top few positions. Theorem 1 demonstrates certain inherent incompatibility between merit-based conditions and group monotonicity when it comes to optimizing welfare. Thus, a compromise must be made between the two, and our choice is the latter one. On one hand, any violation to the merit-based properties is challenging to justify as it undermines creators' perceptions about the value of the matching score metric. If creators discover that highly relevant content can receive lower payoffs or no rewards despite being the most relevant, it can be detrimental to the platform's reputation. On the other hand, while an increase in a creator's matching score $\sigma_i$ would naturally lead to an expected increase in his/her reward, it is generally unnecessary for the total rewards to increase as required by the monotonicity property. In fact, such non-monotonicity is widely observed in free markets, e.g., monopoly vs duopoly markets. For instance, consider a monopoly market with a high-quality producer and a low-quality producer, each catering to their distinct consumer bases. Now suppose the low-quality producer dramatically elevates his/her production quality to transition the market into a duopoly. While such an action would naturally augment the producer's profits, it would concurrently establish intensified competition with the high-quality producer, typically resulting in a marked decline in the latter's profitability. This would subsequently result in a decrease in the two producers' total profit [7, 25]. As will be clear later, our designed rewarding mechanism will lead to similar situations among content creators.

To enhance welfare, it is crucial to incentivize content creators who predominantly target larger user groups to also produce content for smaller groups. However, the monotone property encourages creators to continuously increase their matching scores to a user group, even when those users already have abundant options, resulting in diminishing welfare contributions. To address this, we introduce the class of Backward Rewarding Mechanisms (BRMs). The name of BRM suggests its essential characteristic: the reward for a specific creator-$i$ depends solely on their ranking and the matching scores of creators ranked *lower* than $i$. The formal definition of BRM is provided below:

**Definition 3** (BRM and BRCM). *A Backward Rewarding Mechanism (BRM) $M$ is determined by a sequence of Riemann integrable functions $\{f_i(t) : [0,1] \to \mathbb{R}_{\ge 0}\}_{i=1}^n$, satisfying $f_1(t) \ge \cdots \ge$*

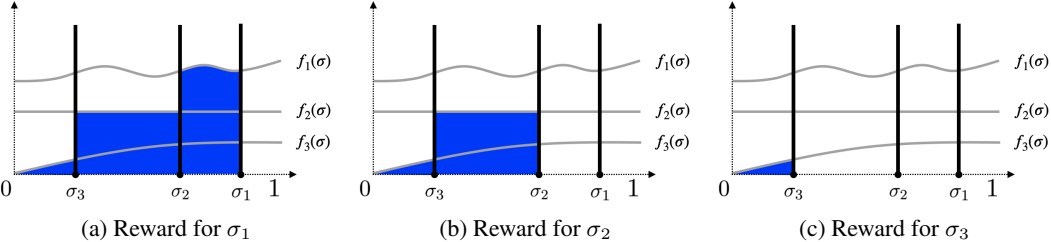

| (a) Reward for $\sigma_1$ | (b) Reward for $\sigma_2$ | (c) Reward for $\sigma_3$ |

Figure 1: An illustration of the BRM parameterized by ordered functions $\{f_1, f_2, f_3\}$. There are 3 creators with matching scores $1 \geq \sigma_1 \geq \sigma_2 \geq \sigma_3 \geq 0$. The area of the blue region in each figure precisely gives the corresponding creator's reward in the BRM.

$f_n(t)$ $\forall t \in [0,1]$, *such that for any matching score sequence* $1 \geq \sigma_1 \geq \cdots \geq \sigma_n \geq 0$*, the reward to any creator $i$ is given by*

$$M(\sigma_i; \sigma_{-i}) = \sum_{k=i}^{n} \int_{\sigma_{k+1}}^{\sigma_k} f_k(t)dt, \tag{5}$$

*where* $\sigma_{n+1} = 0$ *and* $f_1(t) > 0, \forall t \in [0,1]$*. We use* $M[f_1(t), \cdots, f_n(t)]$ *to denote the BRM determined by ordered function sequence* $\{f_i(t)\}_{i=1}^{n}$*.*

*In addition, we identify a sub-class of mechanisms BRCM $\subset$ BRM which includes those $M$ such that $\{f_i(t) \equiv f_i\}$ are a set of constant functions. Any $M \in$ BRCM can be parameterized by an $n$-dimensional vector in the polytope $\mathcal{F} = \{(f_1, \cdots, f_n)|f_1 \geq \cdots \geq f_n \geq 0\}$.*

Figure 1 illustrates an example of BRM with function $\{f_1, f_2, f_3\}$. According to its definition, the blue areas represent the rewards assigned by BRM and we can easily see why BRM preserves the merit-based property. Generally, the function $f_i(\cdot)$ encapsulates the significance of the matching score difference between the $i$-th and $(i+1)$-th ranked content in contributing to the $i$-th ranked creator's reward. The constraint $f_1(t) \geq \cdots \geq f_n(t)$ is necessary to satisfy merit-based properties, as shown in the proof of Proposition 1. The broad class of BRM offers granular control over creator incentives. Meanwhile, the subclass BRCM provides opportunities for parameterized optimization over welfare, which we will discuss in Section 5.2. Additional concrete examples of BRM are illustrated in Appendix 8.3.

## 5.1  Properties of BRM

While the class of BRM might appear abstract at the first glance, one can confirm that it preserves all merit-based properties, making it a natural class of rewarding mechanisms. Nevertheless, in order to secure a better welfare guarantee, the monotonicity is dropped, as characterized in the following:

**Proposition 1.** *Any $M \in$ BRM is merit-based but not necessarily monotone.*

The detailed proof is provided in the Appendix 8.5. Next we establish formal characterizations about the welfare guarantee of BRM. First, we show that any $C^3$ game under BRM possesses a PNE because it is a potential game [16]. A strategic game is called a potential game if there exists a function $P : \prod_i \mathcal{S}_i \rightarrow \mathbb{R}$ such that for any strategy profile $\boldsymbol{s} = (\boldsymbol{s}_1, \cdots, \boldsymbol{s}_n)$, any player-$i$ and strategy $\boldsymbol{s}_i' \in \mathcal{S}_i$, whenever player-$i$ deviates from $\boldsymbol{s}_i$ to $\boldsymbol{s}_i'$, the change of his/her utility function is equal to the change of $P$, i.e.,

$$P(\boldsymbol{s}_i', \boldsymbol{s}_{-i}) - P(\boldsymbol{s}_i, \boldsymbol{s}_{-i}) = u_i(\boldsymbol{s}_i', \boldsymbol{s}_{-i}) - u_i(\boldsymbol{s}_i, \boldsymbol{s}_{-i}).$$

This leads us to the main result of this section:

**Theorem 2.** *Consider any $C^3$ game $\mathcal{G}(\{\mathcal{S}_i\}, \{c_i\}, \mathcal{F}, \sigma, M, \{r_i\})$.*

1. *The $C^3$ game is a potential game under any any mechanism $M \in$ BRM, and thus admits a pure Nash equilibrium (PNE);*

2. *Moreover, if the mechanism $M = M[r_1, \cdots, r_n] \in$ BRCM ($\subset$ BRM), the potential function is precisely the welfare function, i.e., $W(\boldsymbol{s}) = P(\boldsymbol{s}; M)$. Consequently, the always exists a PNE that obtains the optimal welfare.*

The proof is in Appendix 8.6, where we construct its potential function explicitly. According to [16], we also conclude: 1. the maximizers of $P$ are the PNEs of $\mathcal{G}$, and 2. if the evolution of creators' strategic behavior follows a better response dynamics (i.e., in each iteration, an arbitrary creator deviates to a strategy that increases his/her utility), their joint strategy profile converges to a PNE.

Theorem 2 suggests another appealing property of BRM: one can always select an $M$ within BRM to align the potential function with the welfare metric, which can be simply achieved by setting each $f_i$ identical to $r_i$. Consequently, any best response dynamic among creators not only converges to a PNE but also generates a strictly increasing sequence of $W$, thus ensuring at least a local maximizer of $W$. Denote the set of PNEs of $\mathcal{G}$ as $PNE(\mathcal{G})$. When $PNE(\mathcal{G})$ coincides with the global maximizers of its potential function, i.e., $PNE(\mathcal{G}) = \mathrm{argmax}_{\boldsymbol{s}} P(\boldsymbol{s}; M)$, we conclude that any PNE of $\mathcal{G}$ also maximizes the welfare $W$. The following corollary indicates that such an optimistic situation occurs in TvN games, providing a stark contrast to the findings in Theorem 1.

**Corollary 1.** *For any TvN instance $\mathcal{G}$, there exists $M \in BRCM$ such that any PNE $\boldsymbol{s}^* \in PNE(\mathcal{G})$ attains the optimal $W$, i.e.,*

$$\max_{\boldsymbol{s} \in \mathcal{S}} W(\boldsymbol{s}) = W(\boldsymbol{s}^*). \tag{6}$$

The proof is in Appendix 8.7. Despite the promising results presented in Corollary 1, it remains uncertain whether the strong welfare guarantee for TvN can be extended to the entire class of $C^3$. This uncertainty arises because, in general, we only know that $\mathrm{argmax}_{\boldsymbol{s}} P(\boldsymbol{s}; M) \subseteq PNE(\mathcal{G})$. However, [21] noted that the subset of PNEs corresponding to $\mathrm{argmax}_{\boldsymbol{s}} P(\boldsymbol{s}; M)$ in any potential game is robust in the following sense: in an incomplete information relaxation of $\mathcal{G}$, where each creator possesses a private type and must take actions based on their beliefs about other creators' types, they will play the strategies in $\mathrm{argmax}_{\boldsymbol{s}} P(\boldsymbol{s}; M)$ at the Bayesian Nash equilibrium with a probability close to 1. This insight suggests that BRM has the potential to achieve optimal social welfare in real-world scenarios. While we lack a conclusive theoretical determination of whether BRM can attain globally optimal welfare, our empirical study in Section 6 consistently reveals that BRM outperforms baseline mechanisms in $\mathcal{M}^3$ in terms of improving welfare.

## 5.2 Welfare Optimization within BRCM

Theorem 2 suggests that, provided the parameters $\{r_i\}$ are known, the platform can select a mechanism within BRCM with a better welfare guarantee. However, in many practical scenarios, the platform may not have access to the exact values of $\{r_i\}$ but can only evaluate the resulting welfare metric using certain aggregated statistics. This presents a challenge as it may not be analytically feasible to pinpoint the optimal $M$ as suggested by Theorem 2. In these cases, although perfect alignment between the potential function $P$ and social welfare $W$ may not be feasible, we can still find a mechanism that approximates the maximizer of $W$ in creator competition. This leads us to formulate the following bi-level optimization problem:

$$\max_{M \in BRCM} \quad W(\boldsymbol{s}^*(M)) \tag{7}$$

$$\text{s.t.,} \quad \boldsymbol{s}^*(M) = \mathrm{argmax}_{\boldsymbol{s}} P(\boldsymbol{s}; M) \tag{8}$$

In problem (7), the inner optimization (8) is executed by creators: for any given $M$, we have justified that the creators' strategies is very likely to settle at a PNE $\boldsymbol{s}^*(M)$ that corresponds to a maximizer of $P(\boldsymbol{s}; M)$. However, the exact solution to the inner problem is neither analytically solvable by the platform (owing to the combinatorial nature of $P$) nor observable from real-world feedback (due to creators' potentially long feedback cycles). Therefore, we propose to approximate its solution by simulating creators' strategic response sequences (See Appendix 8.8, Algorithm 1), on top of which we solve (7). Algorithm 1 is a variant of better response dynamics, incorporating randomness and practical considerations to more accurately emulate creator behavior, and will be employed as a subroutine in Algorithm 2. We should note that the specifics of the creator response simulator are not critical to our proposed solution: the optimizer can select any equilibrium-finding dynamic to replace our simulator 1, as long as it is believed to better represent creators' responses in reality.

Another challenge of solving (7) lies in the presence of ranking operations in $W$, which makes it non-differentiable in $\boldsymbol{s}$ and renders first-order optimization techniques ineffective. Consequently, we resort to coordinate update and apply finite differences to estimate the ascending direction of $W$ with respect to each $M$ parameterized by $\boldsymbol{f} = (f_1, \cdots, f_n) \in \mathcal{F}$. Our proposed optimization algorithm for solving (7) is presented in Algorithm 2 in Appendix 8.9, and is structured into $L_1$ epochs. At the

beginning of each epoch, the optimizer randomly perturbs the current $M$ along a direction within the feasible polytope and simulates creators' responses for $L_2$ steps using Algorithm 1. Welfare is re-evaluated at the end of this epoch, and the perturbation on $M$ is adopted if it results in a welfare increase.

## 6 Experiments

To validate our theoretical findings and demonstrate the efficacy of Algorithm 2, we simulate the strategic behavior of content creators and compare the evolution of social welfare under various mechanisms. These include Algorithm 2 and several baselines from both the $\mathcal{M}^3$ and BRCM classes.

### 6.1 Specification of Environments

We conduct simulations on game instances $\mathcal{G}(\{\mathcal{S}_i\}, \{c_i\}, \mathcal{F}, \sigma, M, \{r_i\})$ constructed from synthetic data. Results on MovieLens-1m dataset [11] are shown in Appendix 8.10. For the synthetic data, we consider a uniform distribution $\mathcal{F}$ on $\mathcal{X}$ as follows: we fix the embedding dimension $d$ and randomly sample $Y$ cluster centers, denoted as $\mathbf{c}_1, \cdots, \mathbf{c}_Y$, on the unit sphere $\mathbb{S}^{d-1}$. For each center $\mathbf{c}_i$, we generate users belonging to cluster-$i$ by first independently sampling from a Gaussian distribution $\tilde{\boldsymbol{x}} \sim \mathcal{N}(\mathbf{c}_i, v^2 I_d)$, and then normalize it to $\mathbb{S}^{d-1}$, i.e., $\boldsymbol{x} = \tilde{\boldsymbol{x}}/\|\tilde{\boldsymbol{x}}\|_2$. The sizes of the $Y$ user clusters are denoted by a vector $\boldsymbol{z} = (z_1, \cdots, z_Y)$. In this manner, we generate a population $\mathcal{X} = \cup_{i=1}^Y \mathcal{X}_i$ with size $m = \sum_{i=1}^Y z_i$. The number of creators is set to $n = 10$, with action sets $\mathcal{S}_i = \mathbb{S}^{d-1}$. The relevance function $\sigma(\boldsymbol{x}, \boldsymbol{s}) = \frac{1}{2}(\boldsymbol{s}^\top \boldsymbol{x} + 1)$ is the shifted inner product such that its range is exactly $[0, 1]$. $\{r_i\}_{i=1}^n$ is set to $\{\frac{1}{\log_2(2)}, \cdots, \frac{1}{\log_2(5)}, \frac{1}{\log_2(6)}, 0, \cdots, 0\}$. These synthetic datasets simulate situations where content creators compete over a clustered user preference distribution. We consider two types of game instances, denoted $\mathcal{G}_1$ and $\mathcal{G}_2$, distinguished by their cost functions:

1. In $\mathcal{G}_1$, creators have zero cost and their initial strategies are set to the center of the largest user group. This environment models the situation where the social welfare is already trapped at suboptimal due to its unbalanced content distribution. We aim to evaluate which mechanism is most effective in assisting the platform to escape from such a suboptimal state.

2. In $\mathcal{G}_2$, creators have non-trivial cost functions $c_i = 0.5\|\boldsymbol{s}_i - \bar{\boldsymbol{s}}_i\|_2^2$, where the cost center $\bar{\boldsymbol{s}}_i$ is randomly sampled on $\mathbb{S}^{d-1}$. Their initial strategies are set to the corresponding cost centers, i.e., all creators start with strategies that minimize their costs. This environment models a "cold start" situation for creators: they do not have any preference nor knowledge about the user population and gradually learn about the environment under the platform's incentivizing mechanism.

In our experiment, we set $(d, v, Y, m) = (10, 0.3, 8, 52)$ and the cluster sizes $\boldsymbol{z} = (20, 10, 8, 5, 3, 3, 2, 1)$. The 8 clusters are devided into 3 groups $((20), (10, 8), (5, 3, 3, 2, 1))$, namely group-1,2,3, corresponding to the majority, minority, and niche groups.

### 6.2 Algorithm and Baseline Mechanisms

We simulate the welfare curve produced by Algorithm 2 alongside five baseline mechanisms below.

1. BRCM$_{opt}$: This refers to the dynamic mechanism realized by optimization Algorithm 2. The parameters are set to $T = 1000, L_1 = 200, L_2 = 5, \eta_1 = \eta_2 = 0.1, \boldsymbol{f}^{(0)} = (1, 1, 1, 1, 1, 0, \cdots, 0)$.

2. BRCM$^*$: This denotes the theoretically optimal mechanism within BRCM, as indicated by Theorem 2. The corresponding parameters of $M$ are derived based on the knowledge of $\{r_i\}_{i=1}^n$.

3. BRCM$_1$: BRCM$_1 = M[1, \frac{1}{2}, \frac{1}{3}, \frac{1}{4}, \frac{1}{5}, 0, \cdots, 0] \in$ BRCM. This baseline aims to assess the impact of deviation from the theoretically optimal mechanism on the result.

4. $M^3(0)$: This mechanism assigns each content creator a reward equal to the relevance score, i.e., $M(\sigma_i; \sigma_{-i}) = \sigma_i$. It is obvious that this mechanism belongs to the $\mathcal{M}^3$ class and is therefore denoted as $M^3(0)$. Under $M^3(0)$, each creator's strategy does not affect other creators' rewards at all, and thus every creator will be inclined to match the largest user group as much as their cost allows. This mechanism acts as a reference to indicate the worst possible scenario.

5. $M^3(expo.)$: The mechanism based on exposure, defined in Section 4 with $K = 5, \beta = 0.05$.

6. $M^3(enga.)$: The mechanism based on engagement, defined in Section 4 with $K = 5, \beta = 0.05$.

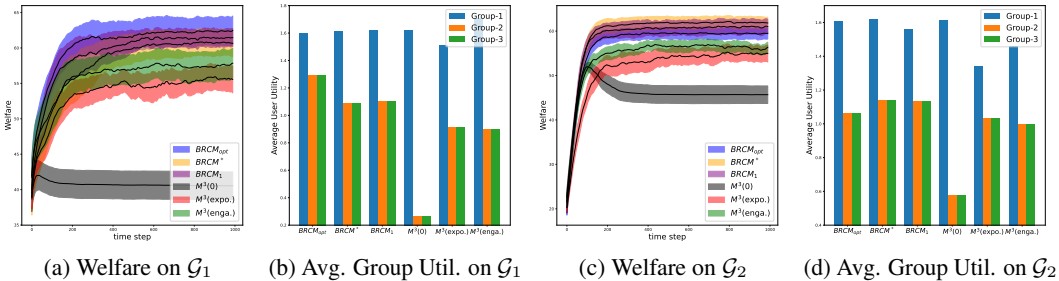

| (a) Welfare on $\mathcal{G}_1$ | (b) Avg. Group Util. on $\mathcal{G}_1$ | (c) Welfare on $\mathcal{G}_2$ | (d) Avg. Group Util. on $\mathcal{G}_2$ |

Figure 2: Social welfare curve and average user utilities per group. Error bars represent half standard deviation range ($0.5\sigma$), and are generated from simulations on 10 randomly sampled game instances.

## 6.3 Results

We let creators play $\mathcal{G}_1$ and $\mathcal{G}_2$ repeatedly under mechanisms specified in Section 6.2 and record the social welfare over $T = 1000$ steps with Algorithm 1 in in Figure 2.

As illustrated in Figure 2a, BRCM family consistently outperformed $\mathcal{M}^3$. As anticipated, $M^3(0)$ does little to enhance social welfare when creators have already primarily focused on the most populous user group. The $M^3(expo.)$ and $M^3(enga.)$ mechanisms demonstrate a notable improvement over $M^3(0)$ as they instigate a competitive environment for creators striving to reach the top-$K$ positions. Nevertheless, they still do not perform as effectively as BRCM$_1$, even though BRCM$_1$'s parameter deviates from the theoretically optimal one. Within the BRCMs, BRCM$_{opt}$ exhibits remarkable performance and even surpasses the theoretically optimal instance BRCM$^*$. One possible explanation for the empirical sub-optimality of BRCM$^*$ is the stochastic nature of creators' response dynamics, which might prevent the convergence to PNE associated with the maximum welfare without sufficient optimization. This observation underscores the importance of Algorithm 2, as it empowers the platform to pinpoint an empirically optimal mechanism in more practical scenarios. As depicted in Figure 2b, the primary source of advantage stems from the increased utility among minority and niche user groups: compared to $M^3(expo.)$ and $M^3(enga.)$, BRCM class results in higher average utility for groups 2 and 3 while preserving overall satisfaction for group-1.

Similar observations can be made for $\mathcal{G}_2$. However, it is worth noting that BRCM$_{opt}$ underperformed slightly in comparison to BRCM$^*$ as shown in Figure 2c. Despite this, the BRCM class of mechanisms continued to significantly surpass those in $\mathcal{M}^3$. Figure 2d further highlights that BRCM mechanisms lead to a more equitable distribution of average user utility across different user groups. Nevertheless, the gap in comparison becomes less pronounced, which is probably due to the existence of costs. Creators burdened with such costs are inherently inclined towards serving specific user groups, making them less susceptible to the influence of platform's incentives.

## 7 Conclusion

Our work reveals an intrinsic limitation of the monotone reward principle, widely used by contemporary online content recommendation platforms to incentivize content creators, in optimizing social welfare. As a rescue, we introduce BRM, a novel class of reward mechanisms with several key advantages. First, BRM ensures a stable equilibrium in content creator competition, thereby fostering a consistent and sustainable content creation environment. Second, BRM can guide content creators' strategic responses towards optimizing social welfare, providing at least a local optimum for any given welfare metric. Finally, BRM offers a parameterized subspace that allows the platform to empirically optimize social welfare, enhancing platform performance dynamically.

For future work, we identify two potential directions. From a theoretical standpoint, it would be intriguing to ascertain whether a stronger welfare guarantee for BRM could be established when the scoring function is equipped with certain simple structures, e.g., dot product. On the empirical side, we look for developments of our suggested mechanism by addressing some practical considerations. For instance, how can we enhance the robustness of BRM to account for the estimation noise in relevance scores? And how can a platform optimize welfare subject to budget constraints? Deeper insights into these questions could significantly enhance our understanding of the rapidly evolving online content ecosystems.

**Acknowledgment**   This work is supported in part by an NSF Award CCF-2303372, IIS-2128019, and IIS-2007492, an Army Research Office Award W911NF-23-1-0030, and an Office of Naval Research Award N00014-23-1-2802.

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

# 8 Supplementary Material

## 8.1 Additional Examples of User Utility Function

Fix any user $x \in \mathcal{F}$, let $\sigma_i = \sigma(s_i; x)$ and for simplicity of notations we assume $\sigma_1 \geq \cdots \geq \sigma_n$. As discussed in Section 3, if the platform presents the top-$K$ ranked content in terms of their relevance quality, the user utility function has the following form:

$$W(s; x) = \sum_{k=1}^{n} r_k \sigma_k, \tag{9}$$

where $\{r_k\}_{k=1}^{n}$ are the user's "attention" over the $k$-th ranked content such that $r_k = 0, \forall k \geq K + 1$. We emphasize that our user utility model given in Eq.(9) is compatible with various matching strategies and here we provide additional examples that incorporate a modified version of the top-$K$ approach, taking into account considerations of advertised content. For instance, considering a scenario where $K = 5$ and the platform intends to promote the content originally ranked at position 6 to position 2 with probability $p \in (0, 1)$. Consequently, the resulting utility function can be expressed as follows:

$$
\begin{aligned}
\tilde{W}_j(s) &= p(r_1\sigma_1 + r_2\sigma_6 + r_3\sigma_2 + r_4\sigma_3 + r_5\sigma_4) + (1-p)(r_1\sigma_1 + r_2\sigma_2 + r_3\sigma_3 + r_4\sigma_4 + r_5\sigma_5) \\
&= r_1\sigma_1 + [pr_3 + (1-p)r_2]\sigma_2 + [pr_4 + (1-p)r_3]\sigma_3 + [pr_5 + (1-p)r_4]\sigma_4 + (1-p)r_5\sigma_5 + pr_2\sigma_6 \\
&\triangleq \sum_{k=1}^{n} \tilde{r}_k \sigma_k.
\end{aligned}
$$

This example shows that user utility function under any position-based perturbation of top-$K$ ranking can be expressed in the form of Eq.(9), and in general the values of $r_k, k > K$ can be non-zero.

## 8.2 Examples of $\mathcal{M}^3$

In this section we formally justify that the two examples given in Section 4 belong to the class of $\mathcal{M}^3$.

1. When the creators' utilities are set to the total content exposure [2, 13, 14], we have $M(\sigma_i; \sigma_{-i}) = \mathbb{I}[i \leq K] \frac{\exp(\beta^{-1}\sigma_i)}{\sum_{j=1}^{K} \exp(\beta^{-1}\sigma_j)}$, with a temperature parameter $\beta > 0$ controlling the spread of rewards. The validity of three merit-based properties are straightforward. In terms of monotonicity, we have $\sum_{i=1}^{n} M(\sigma_i; \sigma_{-i}) = 1$ which is a constant and thus monotone.

2. When the creators' utilities are set to the total user engagement [23], we have $M(\sigma_i; \sigma_{-i}) = \mathbb{I}[i \leq K] \frac{\exp(\beta^{-1}\sigma_i)}{\sum_{j=1}^{K} \exp(\beta^{-1}\sigma_j)} \pi(\sigma_1, \cdots, \sigma_n)$, where $\pi(\sigma_1, \cdots, \sigma_n) = \beta \log\left(\sum_{j=1}^{K} \exp(\beta^{-1}\sigma_j)\right)$. The first two merit-based properties are obvious (Normality and Fairness). In terms of monotonicity, we have $\sum_{i=1}^{n} M(\sigma_i; \sigma_{-i}) = \beta \log\left(\sum_{j=1}^{K} \exp(\beta^{-1}\sigma_j)\right)$ which is monotone in each $\sigma_j$. To verify negative externality, it suffices to show the function $\frac{\log\left(\sum_{j=1}^{K} \exp(\beta^{-1}\sigma_j)\right)}{\sum_{j=1}^{K} \exp(\beta^{-1}\sigma_j)}$ is decreasing in $\sigma_j, \forall j$. Since $\exp(x)$ is increasing in $x$, and function $\frac{\log(t)}{t}$ is decreasing when $t > e$, we conclude that $M$ satisfies negative externality when $n \geq 3$.

## 8.3 Examples of BRM

To get an better intuition of how BRM works, let us consider a special case $M \in \text{BRCM}$ such that $f_1 = \cdots = f_K = 1$ and $f_k = 0, k \geq K+1$. By the definition, any matching score sequence $\sigma_1 \geq \cdots \geq \sigma_n$ will be mapped to a reward sequence of $(\sigma_1 - \sigma_{K+1}, \cdots, \sigma_K - \sigma_{K+1}, 0, \cdots, 0)$. Consequently, the top-$K$ ranked creators will experience a significant reduction in rewards if the $(K + 1)$-th ranked creator increases its matching score. This mechanism can deter an unnecessary concentration of creators on a specific strategy, as when the number of creators with high scores exceeds a certain threshold, even those ranked highly can receive a decreasing reward. This backward rewarding mechanism thus encourages diversity in content creation and mitigates the risk of oversaturation in any particular group of users.

Another notable special case within $\text{BRCM} \in \text{BRM}$ is $M^{SM} = M[1, \frac{1}{2}, \cdots, \frac{1}{n}]$, which coincides with the Shapley mediator proposed in [4]. One key feature of $M^{SM}$ is that for any sequence $1 \geq \sigma_1 \geq \cdots \geq \sigma_n \geq 0$, it holds that $\sum_{i=1}^{n} M^{SM}(\sigma_i; \sigma_{-i}) = \sigma_1 \leq 1$. This implies that the platform can avoid providing explicit incentives and merely implement these rewards as matching probabilities. However, to do so, it must accommodate the possibility of not matching a user with any creators, corresponding to a probability of $1 - \sigma_1$. Furthermore, it does not support the top-$K$ ranking strategy.

A more comprehensive understanding about the construction of BRM can be obtained through the lens of congestion games. As pointed out by [16], every finite potential game is isomorphic to a congestion game. Furthermore, the definition of $M$ as outlined in Eq.(5) can be interpreted as the utility that creator $i$ acquires from the following congestion game:

1. The set of congestible elements are given by the continuum $E = \mathcal{X} \times [0, 1]$, where each element $(\boldsymbol{x}, t) \triangleq \boldsymbol{e} \in E$ corresponds to a user $\boldsymbol{x}$ with satisfaction level $t$.

2. The $n$ players are $n$ content creators.

3. Each creator's pure action $\boldsymbol{s}_i \in \mathcal{S}_i$ can be mapped to a subset of $E$ in the following way: the action $\boldsymbol{s}_i$ determines the matching score $\sigma(\boldsymbol{s}_i; \boldsymbol{x})$ over each $\boldsymbol{x} \in \mathcal{X}$, and then $\boldsymbol{s}_i$ is mapped to a subset $\{(\boldsymbol{x}, t) | \boldsymbol{x} \in \mathcal{X}, t \in [0, \sigma(\boldsymbol{s}_i; \boldsymbol{x})]\} \triangleq S_i \subseteq E$.

4. For each element $\boldsymbol{e}$ and a vector of strategies $(S_1, \cdots, S_n)$, the load of element $\boldsymbol{e}$ is defined as $x_{\boldsymbol{e}} = \#\{i : \boldsymbol{e} \in S_i\}$, i.e., the number of players who occupy $\boldsymbol{e}$.

5. For each element $\boldsymbol{e}$, there is a payoff function $d_{\boldsymbol{e}} : \mathbb{N} \to \mathbb{R}_{\geq 0}$ that only depends on the load of $\boldsymbol{e}$.

6. For any joint strategy $(S_1, \cdots, S_n)$, the utility of player $i$ is given by $\sum_{\boldsymbol{e} \in S_i} d_{\boldsymbol{e}}(x_{\boldsymbol{e}})$, i.e., the sum of reward he/she collects from all occupied elements. For each occupied element $\boldsymbol{e}$, the reward is determined by its "congestion" level $x_{\boldsymbol{e}}$, which is characterized by the payoff function $d_{\boldsymbol{e}}$.

To better understand the constructed congestion game and the utility definition given in Eq.(5), we can consider each element in $E$ (i.e., a user with a particular satisfaction level) as an atomic "resource". Each production strategy adopted by an individual creator can be thought of as occupying a subset of these resources. Given a fixed strategy profile, the load of $\boldsymbol{e} = (\boldsymbol{x}, t)$ is determined by the number of creators who achieve a matching score exceeding $t$ for user $\boldsymbol{x}$, thereby linking the ranking of each creator in the matching score sequence for $\boldsymbol{x}$. Consequently, we can reformulate the utility for a creator who is ranked in the $i$-th position for user $\boldsymbol{x}$ as

$$\sum_{\boldsymbol{e} \in S_i} d_{\boldsymbol{e}}(x_{\boldsymbol{e}}) = \sum_{t \in [0, \sigma(\boldsymbol{s}_i; \boldsymbol{x})]} d_t(x_{\boldsymbol{e}}) = \sum_{k=i}^{n} \sum_{t \in [\sigma(\boldsymbol{s}_{k+1}; \boldsymbol{x}), \sigma(\boldsymbol{s}_k, \boldsymbol{x})]} d_t(x_{\boldsymbol{e}})$$

$$= \sum_{k=i}^{n} \sum_{t \in [\sigma(\boldsymbol{s}_{k+1}; \boldsymbol{x}), \sigma(\boldsymbol{s}_k; \boldsymbol{x})]} d_t(k) \qquad (10)$$

$$\triangleq \sum_{k=i}^{n} \int_{\sigma_{k+1}}^{\sigma_k} f_k(t) dt.$$

Eq.(10) holds because for any resource $\boldsymbol{e} = (\boldsymbol{x}, t)$ such that $t \in [\sigma(\boldsymbol{s}_{k+1}; \boldsymbol{x}), \sigma(\boldsymbol{s}_k; \boldsymbol{x})]$, the load of $\boldsymbol{e}$ is exactly given by $k$. As a result, by letting $f_k(t) = d_t(k)$, we recover the utility function defined in Eq.(5), where the value of function $f_i(t)$ at $t = t_0$ indicates the atomic reward for each creator if his/her strategy covers "resource" $(\boldsymbol{x}, t_0)$, given that there are exactly $i$ creators occupy $(\boldsymbol{x}, t_0)$. This relationship also rationalizes why it is natural to assume that $f_1 \geq \cdots \geq f_n$: as an increase in competition for the same resource from multiple creators should correspondingly reduce the return that can be accrued from that resource.

### 8.4   Proof of Theorem 1

Before showing the proof, we define the following notion of *local* maximizer:

**Definition 4.** *We say $\boldsymbol{s} = (\boldsymbol{s}_1, \cdots, \boldsymbol{s}_n)$ is a* local *maximizer of $W(\boldsymbol{s})$ if for any $i \in [n]$ and any $\boldsymbol{s}_i' \in \mathcal{S}_i$,*

$$W(\boldsymbol{s}_1, \cdots, \boldsymbol{s}_i, \cdots, \boldsymbol{s}_n) \geq W(\boldsymbol{s}_1, \cdots, \boldsymbol{s}_i', \cdots, \boldsymbol{s}_n).$$

*The set of all the local maximizers of $W$ is denoted by $Loc(W)$.*

According to the definition, for any join strategy profile $s \in Loc(W)$, no creator can unilaterally change his/her strategy to increase the value of function $W$. And clearly we have $\arg\max_{s \in S} W(s) \in Loc(W)$. To simplify notation we define $\pi(\sigma_1, \cdots, \sigma_n) = \sum_{i=1}^{n} M(\sigma_i; \sigma_{-i})$. Now we are ready to present the proof of Theorem 1. To avoid complex notations, with a slight abuse of notation we use $M(\sigma_1, \sigma_2, \cdots, \sigma_n)$ to denote $M(\sigma_1; \{\sigma_2, \cdots, \sigma_n\})$ in the following proof.

*Proof.* We start by showing that any TvN game instance with $M \in \mathcal{M}^3$ possesses a unique NE at $s^* = (e_1, \cdots, e_1)$. It suffices to show that:

1. For any joint strategy profile $(s_1, \cdots, s_n)$ in which there are $k < n$ creators occupy $e_1$, there exists a creator who can receive a strict utility gain if she change her strategy to $e_1$.

2. At $s^* = (e_1, \cdots, e_1)$, any player would suffer a utility loss when changing her strategy.

For the first claim, suppose there are $k$ players in $s$ who play $e_1$ and let $i$ be any player who does not play $e_1$. In addition, there are $t \le n - k$ players who play the same strategy as $s_i$. By the definition of $M_3$, we have

$$u_i(s_i; s_{-i}) = 1 \cdot M(\underbrace{1, \cdots, 1}_{t}, \underbrace{0, \cdots, 0}_{n-t}) + (n+1) \cdot M(\underbrace{0, \cdots, 0}_{n-k}, \underbrace{1, \cdots, 1}_{k})$$

$$= 1 \cdot \frac{1}{t} \cdot \pi(\underbrace{1, \cdots, 1}_{t}, \underbrace{0, \cdots, 0}_{n-t}) + (n+1) \cdot 0$$

$$= \frac{1}{t} \cdot \pi(\underbrace{1, \cdots, 1}_{t}, \underbrace{0, \cdots, 0}_{n-t}). \tag{11}$$

If player-$i$ changes her strategy from $s_i$ to $s_i' = e_1$, the new utility would be

$$u_i(s_i'; s_{-i}) = (n+1) \cdot M(\underbrace{1, \cdots, 1}_{k+1}, \underbrace{0, \cdots, 0}_{n-k-1}) + \sum_{j \ne i} 1 \cdot M(0, \cdots)$$

$$= (n+1) \cdot \frac{1}{k+1} \cdot \pi(\underbrace{1, \cdots, 1}_{k+1}, \underbrace{0, \cdots, 0}_{n-k-1}) + 0$$

$$= \frac{n+1}{k+1} \cdot \pi(\underbrace{1, \cdots, 1}_{k+1}, \underbrace{0, \cdots, 0}_{n-k-1}), \tag{12}$$

From Eq.(11) and Eq.(12), $u_i(s_i'; s_{-i}) > u_i(s_i; s_{-i})$ holds if and only if

$$\frac{1}{t} \cdot \pi(\underbrace{1, \cdots, 1}_{t}, \underbrace{0, \cdots, 0}_{n-t}) < \frac{n+1}{k+1} \cdot \pi(\underbrace{1, \cdots, 1}_{k+1}, \underbrace{0, \cdots, 0}_{n-k-1}). \tag{13}$$

And a sufficient condition for Eq.(13) to hold is

$$m = 2n > n - 1 + \max_{0 \le k \le n-1} \left\{ \frac{k+1}{t} \cdot \frac{\pi(\overbrace{1, \cdots, 1}^{t}, \overbrace{0, \cdots, 0}^{n-t})}{\pi(\underbrace{1, \cdots, 1}_{k+1}, \underbrace{0, \cdots, 0}_{n-k-1})} \right\}. \tag{14}$$

Denote $\tilde{\pi}_k = \pi(\underbrace{1, \cdots, 1}_{k}, \underbrace{0, \cdots, 0}_{n-k})$. By the monotonicity of $\pi$, we have $\tilde{\pi}_n \ge \cdots \ge \tilde{\pi}_1 = M(1, 0, \cdots, 0) > 0$. Therefore, the RHS of Eq.(14) is a finite number. Moreover, when $t \le k+1$, we have

$$\frac{k+1}{t} \cdot \frac{\tilde{\pi}_t}{\tilde{\pi}_{k+1}} \le \frac{k+1}{t} \cdot \frac{\tilde{\pi}_{k+1}}{\tilde{\pi}_{k+1}} \le \frac{n-1+1}{1} = n,$$

and when $t > k+1$, based on the negative externality principle of merit-based rewarding mechanism we have

$$\frac{k+1}{t} \cdot \frac{\tilde{\pi}_t}{\tilde{\pi}_{k+1}} = \frac{M(\overbrace{1,\cdots,1}^{t},\overbrace{0,\cdots,0}^{n-t})}{M(\underbrace{1,\cdots,1}_{k+1},\underbrace{0,\cdots,0}_{n-k-1})} \leq 1.$$

Therefore, the RHS of Eq.(14) is strictly less than $2n-1$.

For the second claim, we have

$$u_i(\boldsymbol{s}_i^*; \boldsymbol{s}_{-i}^*) = \frac{n+1}{n}\tilde{\pi}_n,$$

and if player-$i$ changes her strategy from $\boldsymbol{s}_i^* = \boldsymbol{e}_1$ to any $\boldsymbol{s}_i' = \boldsymbol{e}_j, j \neq 1$, her new utility becomes

$$u_i(\boldsymbol{s}_i'; \boldsymbol{s}_{-i}^*) = \tilde{\pi}_1 \leq \tilde{\pi}_n < \frac{n+1}{n}\tilde{\pi}_n = u_i(\boldsymbol{s}_i^*; \boldsymbol{s}_{-i}^*).$$

Therefore, we conclude that $\boldsymbol{s}^* = (\boldsymbol{e}_1, \cdots, \boldsymbol{e}_1)$ is the unique NE of $\mathcal{G}$.

Next we estimate the welfare loss of $\boldsymbol{s}^*$ under any sequence $\{r_i\}_{i=1}^K$. First of all, note that for any $\boldsymbol{s} = (\boldsymbol{s}_1, \cdots, \boldsymbol{s}_n) \in Loc(W)$ and any $2 \leq k \leq n$, if there exists $i \neq j$ such that $\boldsymbol{s}_i = \boldsymbol{s}_j = \boldsymbol{e}_k$, then there must be $k' \in [n]$ such that $\boldsymbol{e}_{k'} \notin \boldsymbol{s}_j$. In this case, $W$ strictly increases if $\boldsymbol{s}_j$ changes to $\boldsymbol{e}_{k'}$. Therefore, for any $2 \leq k \leq n$, the number of elements in $\boldsymbol{s}$ that equal to $\boldsymbol{e}_k$ is either 0 or 1. Let the number of elements in $\boldsymbol{s}$ that equal to $\boldsymbol{e}_1$ be $q$. By definition,

$$W(\boldsymbol{s}) = (n+1)\sum_{i=1}^{\min(K,q)} r_i + (n-q)r_1, \tag{15}$$

$$W(\boldsymbol{s}^*) = (n+1)\sum_{i=1}^{K} r_i.$$

Since $q$ maximizes the RHS of Eq.(15), we have $1 \leq q \leq K$ and $(n+1)r_{q+1} \leq r_1 \leq (n+1)r_q$. Therefore,

$$\frac{\max_{\boldsymbol{s}\in\mathcal{S}} W(\boldsymbol{s})}{W(\boldsymbol{s}^*)} \geq \frac{\min_{\boldsymbol{s}\in Loc(W)} W(\boldsymbol{s})}{W(\boldsymbol{s}^*)}$$

$$\geq \frac{(n+1)\sum_{i=1}^{\min(K,q)} r_i + (n-q)r_1}{(n+1)\sum_{i=1}^{K} r_i}$$

$$\geq \frac{(n+1)\sum_{i=1}^{q} r_i + (n-q)r_1}{(n+1)\sum_{i=1}^{q} r_i + (K-q)r_1}$$

$$= 1 + \frac{(n-K)r_1}{(n+1)\sum_{i=1}^{q} r_i + (K-q)r_1}$$

$$\geq 1 + \frac{(n-K)r_1}{[(n+1)q + (K-q)]r_1}$$

$$\to 1 - \frac{1+1/q}{1+nq/K} + \frac{1}{q}, n \to \infty.$$

Since $1 \leq q \leq K$, we conclude that $\frac{\max_{\boldsymbol{s}\in\mathcal{S}} W(\boldsymbol{s})}{W(\boldsymbol{s}^*)} > 1 - O(\frac{1}{n}) + \frac{1}{K}$ when $n$ is sufficiently large. And therefore we conclude that

$$\frac{W(\boldsymbol{s}^*)}{\max_{\boldsymbol{s}\in\mathcal{S}} W(\boldsymbol{s})} \leq \frac{K}{K+1} + O\left(\frac{1}{n}\right).$$

$\square$

## 8.5 Proof of Proposition 1

*Proof.* To prove that any $M \in$ BRM is merit-based, we need to verify the following by definition:

1. $M(0; \sigma_{-i}) = \int_0^0 f_n(t)dt = 0$, $M(1; \{0, \cdots, 0\}) = \int_0^1 f_1(t)dt > 0$.

2. $M(\sigma_i; \sigma_{-i}) - M(\sigma_j; \sigma_{-j}) = \sum_{k=j}^{i-1} \int_{\sigma_{k+1}}^{\sigma_k} f_k(t)dt \geq 0$.

3. for any $\{\sigma_j\}_{j=1}^n, \{\sigma_j'\}_{j=1}^n$ such that $\sigma_{-i} \preccurlyeq \sigma_{-i}'$, we can transform $\{\sigma_j\}_{j=1}^n$ to $\{\sigma_j'\}_{j=1}^n$ by taking finite steps of the following operations: 1. increase a certain value of $\sigma_j, j \neq i$ to $\tilde{\sigma}_j$ and it does not change the order of the current sequence; 2. increase a certain value of $\sigma_j, j \neq i$ to $\tilde{\sigma}_j$, and $\sigma_i$'s ranking position decreases after this change. We will show that after each operation the value of $M(\sigma_i, \cdot)$ under the perturbed sequence does not increase.

Let the perturbed sequence be $\tilde{\sigma}$. For the first type of operation, if $j < i$, we have $M(\sigma_i; \tilde{\sigma}_{-i}) = M(\sigma_i; \sigma_{-i})$. If $j > i$, we have

$$M(\sigma_i; \tilde{\sigma}_{-i}) - M(\sigma_i; \sigma_{-i}) = \int_{\tilde{\sigma}_j}^{\sigma_{j-1}} f_{j-1}(t)dt + \int_{\sigma_{j+1}}^{\tilde{\sigma}_j} f_j(t)dt - \int_{\sigma_j}^{\sigma_{j-1}} f_{j-1}(t)dt - \int_{\sigma_{j+1}}^{\sigma_j} f_j(t)dt$$

$$= \int_{\sigma_j}^{\tilde{\sigma}_j} (f_j - f_{j-1})(t)dt \leq 0.$$

For the second type of operation, with out loss of generality let's assume $\sigma_{i+1}$ has increased to $\tilde{\sigma}_{i+1}$ such that $\sigma_i \leq \tilde{\sigma}_{i+1} \leq \sigma_{i-1}$. In this case we have

$$M(\sigma_i; \tilde{\sigma}_{-i}) - M(\sigma_i; \sigma_{-i}) = \int_{\sigma_{i+2}}^{\sigma_i} f_{i+1}(t)dt - \int_{\sigma_{i+1}}^{\sigma_i} f_i(t)dt - \int_{\sigma_{i+2}}^{\sigma_{i+1}} f_{i+1}(t)dt$$

$$= \int_{\sigma_{i+1}}^{\sigma_i} (f_{i+1} - f_i)(t)dt \leq 0.$$

Therefore, $M$ is merit-based. On the other hand, there exist instances in BRM that are not monotone. For example, if we let $f_1(t) = 1$ and $f_k(t) = 0, \forall k \geq 2$. Then we have

$$M(1, 0, 0, \cdots, 0) = \int_0^1 f_1(t)dt > 0,$$

$$M(1, 1, 0, \cdots, 0) = \int_0^0 f_1(t)dt + \int_0^1 f_2(t)dt = 0.$$

As a result, $\pi(1, 0, 0, \cdots, 0) > 0 = \pi(1, 1, 0, \cdots, 0)$, which violates monotonicity. □

## 8.6 Proof of Theorem 2

*Proof.* For the first claim, consider the potential function of the following form:

$$P(s) = \mathbb{E}_{x \in \mathcal{F}} \left[ \sum_{i=1}^n \int_0^{\sigma_{l(i)}(x)} f_i(t)dt \right] - \sum_{i=1}^n c_i(s_i),$$

where $\sigma_i(x) = \sigma(s_i; x)$ and $\{l(i)\}_{i=1}^n$ is a permutation such that $\sigma_{l(1)}(x) \geq \sigma_{l(2)}(x) \geq \cdots \geq \sigma_{l(n)}(x)$.

By the definition of potential games, we need to verify that for any set of functions $\{f_i\}$ and a strategy pair $s_i, s_i' \in \mathcal{S}_i$ for player-$i$, it holds that

$$u_i(s_i', s_{-i}) - u_i(s_i, s_{-i}) = P(s_i', s_{-i}) - P(s_i, s_{-i}). \tag{16}$$

For any user $x \in \mathcal{F}$, let $\sigma_i = \sigma(s_i; x_j), \sigma_i' = \sigma(s_i'; x), \forall i \in [n]$. It suffices to show that

$$M(\sigma_i; \sigma_{-i}) - M(\sigma_i'; \sigma_{-i}) = \sum_{i=1}^n \int_0^{\sigma_{l(i)}} f_i(t)dt - \sum_{i=1}^n \int_0^{\sigma_{l(i)}'} f_i(t)dt. \tag{17}$$

Since the expectation of Eq.(17) over $\boldsymbol{x} \in \mathcal{F}$ yields Eq.(16), we focus on the verification of Eq.(17). With out loss of generality, we also assume $\sigma_1 \geq \cdots \geq \sigma_{i'} \geq \cdots \geq \sigma_i \geq \cdots \geq \sigma_n$. After player-$i$ changes her strategy from $\boldsymbol{s}_i$ to $\boldsymbol{s}_i'$, the relevance ranking increases from $i$ to $i'$, i.e., $\sigma_1 \geq \cdots \geq \sigma_{i'-1} \geq \sigma_i' \geq \sigma_{i'} \geq \cdots \geq \sigma_n$.

Therefore, we have

$$\text{LHS of Eq.(17)} = \int_{\sigma_{i'}}^{\sigma_i'} f_{i'}(t)dt + \sum_{k=i'+1}^{n} \int_{\sigma_{k-1}}^{\sigma_k} f_k(t)dt, \tag{18}$$

$$\text{RHS of Eq.(17)} = \sum_{k=1}^{n} \int_0^{\sigma_k} f_k(t)dt - \left( \sum_{k=1}^{i'-1} \int_0^{\sigma_k} f_k(t)dt + \int_0^{\sigma_i'} f_{i'}(t)dt + \sum_{k=i'+1}^{n} \int_0^{\sigma_{k-1}} f_k(t)dt \right)$$

$$= \int_0^{\sigma_{i'}} f_{i'}(t)dt + \sum_{k=i'+1}^{n} \int_0^{\sigma_k} f_k(t)dt - \int_0^{\sigma_i'} f_{i'}(t)dt - \sum_{k=i'+1}^{n} \int_0^{\sigma_{k-1}} f_k(t)dt$$

$$= \int_{\sigma_{i'}}^{\sigma_i'} f_{i'}(t)dt + \sum_{k=i'+1}^{n} \int_{\sigma_{k-1}}^{\sigma_k} f_k(t)dt.$$

Hence, Eq.(17) holds for any $j$ which completes the proof.

For the second claim, we can verify that when $f_i = r_i, \forall i$,

$$P(\boldsymbol{s}) = \mathbb{E}_{\boldsymbol{x} \in \mathcal{F}} \left[ \sum_{i=1}^{n} \int_0^{\sigma_{l(i)}(\boldsymbol{x})} f_i(t)dt \right] - \sum_{i=1}^{n} c_i(\boldsymbol{s}_i)$$

$$= \mathbb{E}_{\boldsymbol{x} \in \mathcal{F}} \left[ \sum_{i=1}^{n} r_i \sigma_{l(i)}(\boldsymbol{x}) \right] - \sum_{i=1}^{n} c_i(\boldsymbol{s}_i)$$

$$= \mathbb{E}_{\boldsymbol{x} \in \mathcal{F}} \left[ W(\boldsymbol{s}; \boldsymbol{x}) \right] - \sum_{i=1}^{n} c_i(\boldsymbol{s}_i)$$

$$= W(\boldsymbol{s}).$$

$\square$

## 8.7 Proof of Corollary 1

*Proof.* We show that any TvN game instance $\mathcal{G}$ with $M = M[r_1, \cdots, r_K, 0, \cdots, 0] \in$ BRCM possesses a unique NE $\boldsymbol{s}^*$ which maximizes $W(\boldsymbol{s})$. From Theorem 2 we know that under $M$, $\mathcal{G}$ is a potential game and its potential function $P$ is identical to its welfare function $W$. Therefore, any PNE of $\mathcal{G}$ belongs to $Loc(W)$. Next we show that all elements in $Loc(W)$ yield the same value of $W$, thus any PNE of $\mathcal{G}$ maximizes social welfare $W$.

First of all, note that for any $\boldsymbol{s} = (\boldsymbol{s}_1, \cdots, \boldsymbol{s}_n) \in Loc(W)$ and any $2 \leq k \leq n$, if there exists $i \neq j$ such that $\boldsymbol{s}_i = \boldsymbol{s}_j = \boldsymbol{e}_k$, then there must exist $k' \in [n]$ such that $\boldsymbol{e}_{k'} \notin \boldsymbol{s}_j$. In this case, $W$ strictly increases if $\boldsymbol{s}_j$ changes strategy to $\boldsymbol{e}_{k'}$. Therefore, for any $2 \leq k \leq n$, the number of elements in $\boldsymbol{s}$ that equal to $\boldsymbol{e}_k$ is either 0 or 1. Let the number of elements in $\boldsymbol{s}$ that equal to $\boldsymbol{e}_1$ be $q$. By definition, the welfare function writes

$$W(\boldsymbol{s}) = (n+1) \sum_{i=1}^{\min(K,q)} r_i + (n-q)r_1. \tag{19}$$

It is clear that the $q$ that maximizes Eq.(19) satisfies $1 \leq q \leq K$ and $(n+1)r_{q+1} \leq r_1 \leq (n+1)r_q$, and all such $q$ yields the same objective value of $W$. Therefore, we conclude that any PNE of $\mathcal{G}$ attains the optimal social welfare $W$. $\square$

## 8.8 Content Creator Response Simulator

Algorithm 1 functions as follows: at each step, a random creator $i$ selects a random improvement direction $\boldsymbol{g}_i$. If creator $i$ discovers that adjusting her strategy in this direction yields a higher utility,

---

**Algorithm 1 (simStra)** Simulate content creators' strategy evolving dynamic

---

**Input:** Time horizon $T$, learning rate $\eta$, utility function strategy set $(u_i(\boldsymbol{s}), \mathcal{S}_i)$ for each player, current mechanism $M[\boldsymbol{f}]$ parameterized by $\boldsymbol{f}$.

**Initialization:** Initial strategy profile $\boldsymbol{s}^{(0)} = (\boldsymbol{s}_1^{(0)}, \cdots, \boldsymbol{s}_n^{(0)})$.

**for** $t = 0$ **to** $T - 1$ **do**

    Generate $i \in [n]$ and $\boldsymbol{g}_i \in \mathbb{S}^{d-1}$ uniformly at random.

    **if** $u_i(\boldsymbol{s}_i^{(t)} + \eta\boldsymbol{g}_i, \boldsymbol{s}_{-i}^{(t)}) \geq u_i(\boldsymbol{s}^{(t)})$ **then**

        $\boldsymbol{s}_i^{(t+\frac{1}{2})} = \boldsymbol{s}_i^{(t)} + \eta\boldsymbol{g}_i$.

        Find $\boldsymbol{s}_i^{(t+1)}$ as the projection of $\boldsymbol{s}_i^{(t+\frac{1}{2})}$ in $\mathcal{S}_i$.

    **else**

        $\boldsymbol{s}_i^{(t+1)} = \boldsymbol{s}_i^{(t)}$

**Output:** $\boldsymbol{s}^{(T)}$.

---

she updates her strategy along $\boldsymbol{g}_i$; otherwise, she retains her current strategy. This approach is designed to more closely mimic real-world scenarios where content creators may not have full access to their utility functions, but instead have to perceive them as black boxes. While they may aim to optimize their responses to the current incentive mechanism, identifying a new strategy that definitely increases their utilities can be challenging. Therefore, we model their strategy evolution as a trial-and-exploration process. We should note that the specifics of the simulator are not critical to our proposed solution: the optimizer can select any equilibrium-finding dynamic to replace our Algorithm 1, as long as it is believed to better represent creators' responses in reality.

## 8.9 Optimization Algorithm

Our proposed welfare optimization algorithm is presented in Algorithm 2, which is organized into $L_1$ epochs. Each epoch starts with a random perturbation of the current mechanism $M$ within the feasible polytope and conducts simulations of creators' responses for $L_2$ steps using the simulator specified in Algorithm 1. Welfare is reassessed at the conclusion of each epoch, and the perturbation applied to $M$ is adopted if it yields an increase in welfare.

---

**Algorithm 2** Optimize $W$ in BRCM

---

**Input:** Time horizon $T = L_1 L_2$, learning rate $\eta_1, \eta_2$, $(u_i(\boldsymbol{s}), \mathcal{S}_i)$ for each creator.

**Initialization:** Unit basis $\{\boldsymbol{e}_i\}_{i=1}^n$ in $\mathbb{R}^n$, initial strategy profile $\boldsymbol{s}^{(0)} = (\boldsymbol{s}_1^{(0)}, \cdots, \boldsymbol{s}_n^{(0)})$, initial parameter $\boldsymbol{f}^{(0)} = (f_1^{(0)}, \cdots, f_n^{(0)}) \in \mathcal{F}$ and mechanism $M[\boldsymbol{f}^{(0)}]$.

**for** $t = 0$ **to** $L_1 - 1$ **do**

    Generate $i \in [n]$ and $\boldsymbol{g}_i \in \{-\boldsymbol{e}_i, \boldsymbol{e}_i\}$ uniformly at random.

    Update $\boldsymbol{f}_i^{(t+\frac{1}{2})}$ as the projection of $\boldsymbol{f}_i^{(t)} + \eta_1\boldsymbol{g}_i$ on $\mathcal{F}$.

    Simulate $\boldsymbol{s}^{(t+1)} =$ **simStra**$(\boldsymbol{s}^{(t)}; L_2, \eta_2, \{\boldsymbol{u}_i, \mathcal{S}_i\}_{i=1}^n, M[\boldsymbol{f}^{(t+\frac{1}{2})}])$. // Implemented by Algo. 1

    **if** $W(\boldsymbol{s}^{(t+1)}) > W(\boldsymbol{s}^{(t)})$ **then**

        $\boldsymbol{f}^{(t+1)} = \boldsymbol{f}^{(t+\frac{1}{2})}$.

    **else**

        $\boldsymbol{f}^{(t+1)} = \boldsymbol{f}^{(t)}$.

---

## 8.10 Additional Experiments

• **Experiments using MovieLens-1m** We use deep matrix factorization [8] to train user and movie embeddings predicting movie ratings from 1 to 5 and use them to construct the user population $\mathcal{X}$ and creators' strategy set $\{\mathcal{S}_i\}$. The dataset contains 6040 users and 3883 movies in total, and the embedding dimension is set to $d = 32$. To validate the quality of the trained representation, we first performed a 5-fold cross-validation and obtain an averaged RMSE $= 0.739$ on the test sets, then train the user/item embeddings with the complete dataset.

To construct a more challenging environment for creators, we avoid using movies that are excessively popular and highly rated or users who are overly active and give high ratings to most movies. This

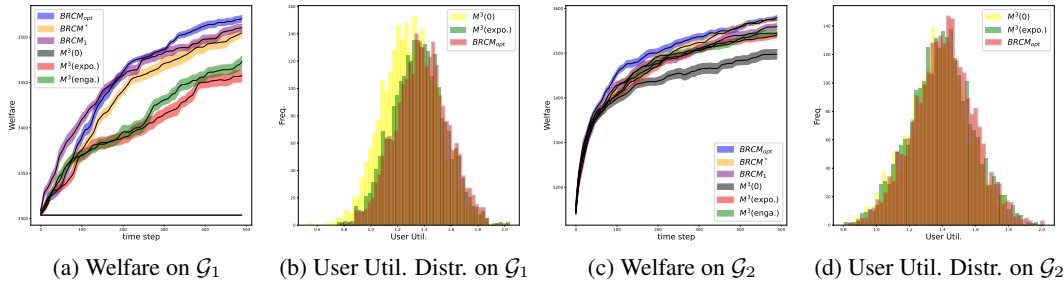

| (a) Welfare on $\mathcal{G}_1$ | (b) User Util. Distr. on $\mathcal{G}_1$ | (c) Welfare on $\mathcal{G}_2$ | (d) User Util. Distr. on $\mathcal{G}_2$ |

Figure 3: Social welfare curve and average user utilities under two different environments. Error bars represent 0.2 standard deviation range, and they are generated from 10 independent runs. Game instances are generated from MovieLens-1m dataset.

ensures that the strategy of "producing popular content for the majority of active users" does not become a dominant strategy under any rewarding mechanism. Thus, we filtered out users and movies who have more than 500 predicted ratings higher than 4. After the filtering, we have a user population with size $|\mathcal{X}| = 2550$ and movie set of size 1783. The user distribution are set to uniform distribution over $\mathcal{X}$, and the remaining movies become the action set $\{\mathcal{S}_i\}$ for each creator-$i$ of $n = 10$ creators. To normalize the relevance score to $[0, 1]$, we set $\sigma(\boldsymbol{s}, \boldsymbol{x}) = \text{clip}(\langle \boldsymbol{s}, \boldsymbol{x} \rangle / 2.5 - 1, 0, 1)$. $\{r_i\}_{i=1}^n$ is set to $\{\frac{1}{\log_2(2)}, \frac{1}{\log_2(3)}, \frac{1}{\log_2(4)}, \frac{1}{\log_2(5)}, \frac{1}{\log_2(6)}, 0, \cdots, 0\}$. We also consider two types of game instances, namely $\mathcal{G}_1$ and $\mathcal{G}_2$, as we elaborated on in Section 6.1. Specifically, in $\mathcal{G}_1$ creators' initial strategies are set to the most popular movie among all users (i.e., the movie that enjoys the highest average rating among $\mathcal{X}$) and the cost functions are set to be zero. In $\mathcal{G}_2$, we set creators' cost functions to $c_i = 10 \|\boldsymbol{s}_i - \bar{\boldsymbol{s}}_i\|_2^2$ and let creator $i$ start at the cost center $\bar{\boldsymbol{s}}_i$. $\{\bar{\boldsymbol{s}}_i\}_{i=1}^n$ are sampled at random from all the movies.

For each baseline in Section 6.2, we plot the welfare curve over $T = 500$ steps using Algorithm 1 and also the average user utility distribution at the end of simulations. The parameters of Algorithm 2 are set to $L_1 = 100, L_2 = 5, \eta_1 = 0.5, \eta_2 = 0.1, \boldsymbol{f}^{(0)} = (1, 1, 1, 1, 1, 0, \cdots, 0)$. The results are presented in Figure 3.

The new results obtained reinforce the findings presented in Section 6. In both the $\mathcal{G}_1$ and $\mathcal{G}_2$ environments, the BRCM family continues to outperform $\mathcal{M}^3$ overall. Specifically, $\text{BRCM}_{opt}$, $\text{BRCM}_1$, and $\text{BRCM}^*$ consistently demonstrate strong performance in social welfare, highlighting the robustness of BRCM across different environments. When creators initially adopt the most popular strategy in $\mathcal{G}_1$, $M^3(0)$ does not yield any improvement since no creator would change their strategy in such a situation under $M^3(0)$. In the case of $\mathcal{G}_2$, the advantage of BRCM over $\mathcal{M}^3$ diminishes slightly, which aligns with our observations from the synthetic dataset. The main reason is that the cost function discourages creators to deviate from their default strategies. Additionally, Figure 3b provides further evidence that the welfare gain achieved by BRCM arises from enhanced utility for a wider range of users.

