# OpenReview forum: "Rethinking Incentives in Recommender Systems: Are Monotone Rewards Always Beneficial?"
_NeurIPS.cc/2023/Conference — NeurIPS 2023 poster_

### Official Review · Reviewer_oV5f · 2023-07-04

**Soundness:** 3 good
**Presentation:** 2 fair
**Contribution:** 2 fair
**Rating:** 5
**Confidence:** 3

**Summary:**

The development of online media referral platforms has provided a source of income for media content creators, and the incentive strategies of the platforms may influence the creators' creative trends. The incentive model that tends to reward may also invariably encourage creators to over-serve the majority user group, and the niche groups will be increasingly underserved. In order to solve these problems, this paper designs a backward incentive mechanism, which induces creators' behavior through the game structure to dynamically optimize the creation model and maximize social welfare. And its advantages are verified by simulation experiments.

**Strengths:**

To address the current problem that rewards mechanisms in online content recommendation platforms affect creators' production choices, the platform's content distribution and social welfare. In this paper, we design a reverse reward mechanism, which can guide content creators to optimize their creation strategies and provide locally optimal results for a given welfare metric. This avoids most creators to generate a large amount of homogeneous content that caters to the majority group for the sake of rewards.
1. originality
In this paper, we provide a reverse reward mechanism to address the problem that current reward mechanisms may encourage creators to concentrate on the mass range, thus leaving niche users unserved. The superiority of this mechanism, which is performance-based but discards monotonicity, has been proven in empirical studies, and the approach in this paper is well original.
2. quality
The research problem of this paper is apparent, the method is introduced in detail, the logic is clear and the experiment is reliable. This paper has good quality.
3. clarity
This paper clearly defines the research problem, and introduces the effectiveness of the method from the theoretical and simulation experiments in detail. The overall clarity is good.
4. significance
In this paper, it designs a reverse incentive strategy to cope with the incentive drift caused by the profit-oriented incentive strategy in online content recommendation platforms, so as to reduce the undesirable incentives that cause a large number of creators to ignore the niche groups and leave them unserved. The research in this paper has significant implications.


**Weaknesses:**

This paper’s related work on the content of the current study is weak, there are few references, and there are only a few works in the past five years, which is a bit difficult to explain the novelty of the current method.

**Questions:**

This paper designs an incentive mechanism for creators in online content recommendation platforms, and its reverse incentive mechanism establishes a gaming competition environment among content creators to maximize social welfare. However, this paper has the following problems:
1. the limitations of this paper are not declared;
2. whether the order of content in sections 3 and 4 can be switched, which seems to be a bit unreasonable at present;
3. in the experimental part, the data used in this paper and some experimental details are not clear descriptions. For example, synthetic data are used in this paper, but the details of the synthetic data are not shown, and for the results in Figure 1, it is not clear from the content whether synthetic data or MovieLens-1m dataset is used;
4. some references are too old and could be replaced with newer references, research from the last five years could be added appropriately, and the paper is not standardized in its citation of references;


**Limitations:**

This paper seems to have no clear statement on the limitations of the research. I have a question as to whether the creators in the current study only considered those who create for profit, and whether they considered those who create for interest. The current study seems to consider creators who create for profit, but it does not seem to be stated in the paper.

---

> ### Author Rebuttal · Authors · 2023-08-09
>
> **Q1: Discussion of limitations** One limitation of our work pertains to the focus of our model, which mainly addresses the challenges encountered by those "strong platforms" which dominate content distribution (e.g., Instagram reels & TikTok). On these platforms, creators’ impressions and rewards are almost enforced by the platform’s design, rendering our game structure and modeling assumptions readily justifiable. However, it should be noted that other content platforms, such as Yelp and Amazon, operate differently, wherein creators’ impressions and rewards may originate from other sources (e.g., user searches), so our assumption does not strictly hold.  Although we believe the insight revealed from our result still generalizes, it requires significantly additional effort to substantiate our claims. We will clarify the limitations in our revision.
>
> **Q2: Switching Section 3 and 4** We appreciate the reviewer’s suggestion, but after some careful consideration, we still believe the current order is better for presentation since we need to first present the framework of our model in Section 3 before describing the class of merit-based monotone mechanisms in Section 4, where we have to use the notations and concepts introduced earlier in Section 3.
>
> **Q3: Description of dataset** The dataset description for the experiment is provided in Section 6.1. And more details about the dataset and our simulation settings can be found in the supplementary document, Appendix 8.10.
>
> **Q4: References** Thanks for pointing this out. We will for sure address these reference issues in the final version.

---

> > ### Comment · Reviewer_oV5f · 2023-08-15
> >
> > Thanks for the response.

---

> > > ### Author Response · Authors · 2023-08-16
> > > **Response**
> > >
> > > We hope that our clarifications have satisfactorily addressed the reviewer’s questions.  Any further feedback or insights that could increase the reviewer’s evaluation of our work are warmly welcomed. Such insights would be invaluable in improving both the quality and impact of our study. Thank you!

---

### Official Review · Reviewer_88Nz · 2023-07-05

**Soundness:** 3 good
**Presentation:** 3 good
**Contribution:** 3 good
**Rating:** 8
**Confidence:** 4

**Summary:**

The authors study strategic content creation in recommendation systems, focusing on the induced game's social welfare. The authors assume that the provider's rewards are entirely determined by the platform's payments, not clicks/engagements. This separation between the ranked results and the creators' incentives facilitates analyzing an expressive game in which the recommender system recommends a list of items (most prior work considers one item) with position bias. Notably, the authors assume that the rewarding mechanism is a mapping from a vector of relevance scores to reward vectors, i.e., $[0,1]^n \rightarrow [0,1]^n$.

The paper considers two classes of merit-based mechanisms, monotone and BRM (and also the BRCM subclass). It showcases evidence against monotone mechanisms, highlighting that in a particular class of games (TvN), the POA of any monotone mechanism is at most $\frac{K}{K+1}$, where $K$ is the length of the list, plus a small $\frac{1}{n}$ factor. Later, for BRCM mechanisms, the authors show that the welfare function is the potential function; thus, the global optimum of the welfare is a PNE (despite the POA could still be  $<1$ for some mechanisms).

Finally, the authors describe how to optimize over BRCM mechanisms in the presence of data, and conduct synthetic and semi-synthetic experiments to demonstrate their approach.


**Strengths:**

1.	The paper deals with a timely and important topic, and well-connects to previous literature.
2.	The paper non-trivially extends previous literature, suggesting new theory and experimental validation.
3.	The optimization problem the paper suggests is exciting and new to this literature.


**Weaknesses:**

1.	Due to the abundance of mathematical objects and notations, the paper is non-trivial to follow. Perhaps this is inevitable, but I see this as a weakness.
2.	The case against monotonicity focuses on a relatively small class of TvN games tailored to the authors' argument. Arguing against monotonicity in (more) general games would be much more convincing.
3.	The empirical evaluation lacks proper benchmarks.


**Questions:**

1.	An assumption of the model is that impressions/clicks of users and creators' rewards are separated and orthogonal. In reality, the utility function of creators is not always "set" by the platform (as argued in 168,170), but rather determined by the interaction the platform allows. For instance, in Medium, blog traffic means more ads, regardless of relevance. Can the platform decide that a blog is irrelevant and thus, despite its heavy traffic, not share the ad revenue with the blog's creator? As written in my summary, I see this separation as the element facilitating a much more expressive game than those considered in prior work. So my question is: Can the authors justify this modeling assumption?
2.	Relating to the previous point, something disturbs me in the explanation in Lines 241-250. Say the $K$ contents creators attract impressions, clicks, etc., and get high rewards. Then, the $K+1$ creators improves their content's relevance, and suddenly the $K$ selected creators suddenly get almost none. I think that this is hard to digest and probably will be deemed unfair from the creators' perspective. Could the authors justify why BRCM mechanisms benefit creators? After all, there are other unfair mechanisms that dictate the content each creator creates, which receives optimal welfare.
3.	The paper advocates using BRCM mechanisms and not monotone ones. Noticeably, the intersection between these classes is not empty. But Theorem 1 suggests that the optimal BRCM mechanism that exists due to Corollary 1 is *not* monotone. This brings the question of how natural this mechanism is (the one guaranteed by Corollary 1). Do we want to sacrifice simplicity (and arguably fairness) for a proportion of $\frac{1}{K+1} of the welfare? Since advocating for BRCM is not equivalent to advocating against monotonicity (as the intersection is non-empty), I think the reader could benefit from a more elaborated discussion. Ideally, this paper would have presented evidence beyond social welfare in TvN games, but perhaps this is saved for future work.
4.	The welfare function comprises the user welfare, the creators rewards and costs, and the mechanism's payments. But the mechanism's payments are precisely the creator rewards, so they cancel. Consequently, by offering unbounded rewards the platform does not change the welfare, but obviously this will never happen in real life. What part of the modeling forbids that? This seems like a loophole in the modeling. Could the authors justify this welfare modeling? Even if we care about the welfare function defined in this paper, one would expect the platform to minimize its payments for any given welfare level. Could this be injected into the model?
5.	In the experimental part, the authors compare BRCM mechanisms with monotone mechanisms. However, for the monotone mechanisms, the authors do not optimize over the hyperparameter $\beta$. I believe an apples-to-apples comparison requires optimizing for both classes of mechanisms. Why didn't the authors do that in their analysis? Additionally, what would be other monotone baselines?
6.	In Figure 1(a), the optimized BRCM scores better than the optimal BRCM. How could that be possible?

Minor:
•	23: “benefits\cite{}” -> add space

•	Lines 139-140 say "Our objective is to design mechanisms M that: 1. guarantee the existence of PNE, thereby ensuring a stable outcome, and 2. maximize social welfare at the PNE." While Line 135 says that this is not the focus of [1,12,13], I think that [1] primarily addresses the same objectives.

•	155: the parameter $n$ seems to be out of context, as it doesn't appear in the $M^3$ notation.

•	210: ".." -> "."

•	254: $BRCM \in BRM$ -> $BRCM \subset BRM$



**Limitations:**

No limitations

---

> ### Author Rebuttal · Authors · 2023-08-09
>
> **Negative result being restrictive** We admit that theoretically demonstrating the limitation of monotonicity in general is challenging, and we identify it as an intriguing future work. In the meantime, our experiment result empirically suggests a constant fraction of welfare loss beyond the class of TvN games: Figure 1 (a, c) indicates that when the user population is well separated, the welfare gap induced by representatives from BRCM and $M^3$ can be approximately 20% (55+ v.s. 65+ in (a) and 50+ v.s. 60+ in (b)) when $K=5$.
>
> **Q1: Justification of modeling assumption** Our modeling assumption mainly focuses on those "strong platforms" which dominate the content distribution (e.g., Instagram reels & TikTok). On these platforms, the correlation between creators’ impressions and rewards is almost “enforced” by the platform’s design, rendering our game structure and modeling assumptions readily justifiable. However, platforms such as Medium and OnlyFans are not precisely the content recommendation platforms we consider in our model since the creators’ impressions and rewards may originate from other sources (e.g., user searches and subscriptions). We appreciate the reviewer for bringing up this concern, and we will include this point in the discussion of limitations.
>
> We should also clarify that for those "strong platforms", we did not assume that the impression/clicks from users and the rewards from the platform are orthogonal. In Line 102-108, we explain how these two are connected: once the platform decides the expected reward for each creator using a mechanism $M$, it can implement it by setting a traffic allocation scheme $p$ and a post allocation reward $R$. More importantly, the platform does have the flexibility to make the impressions/clicks from users and the rewards from the platform positively correlated. The easiest way to do so is to allocate traffic proportional to the output of $M$ by setting $R=1$. In this case, due to the fairness property of BRM (see Line 151), the platform can always guarantee a creator with a higher relevance score receives a larger portion of the traffic.
>
> **Q2: Why BRM has benefits** The rationale behind the example of BRCM[1,...,1,0,...,0] is that the marginal contribution of the $K$-th creator to the user satisfaction becomes smaller when the $K+1$-th creator gets a similar score since even if the $K$-th creator lowers her content quality, the satisfaction of this user is not severally impacted as the $K+1$-th creator can fill in the $K$-th position. Therefore, BRM benefits creators by preventing the rat race among them: everyone's utility would drastically decrease as the competition over a user interest group becomes unnecessarily intense. Such signals can thus help creators to identify and reroute to underserved audiences for larger potential rewards. This bears semblance to the analogy of traffic congestion, where an individual's utility declines when their chosen route becomes congested, prompting a shift towards an alternative path with less congestion.
>
> If the platform worries that this mechanism might be intricate for creators to digest, it can simply inform creators that the reward from a user impression is proportional to the creator’s marginal contribution to this user’s expected satisfaction, which means the creator will get a lower reward if lots of high quality but similar content already overload this user.
>
> **Q3: Simplicity of BRCM** Thanks for pointing out that the intersection between M3 and BRCM is not empty, and we will add corresponding discussions in the revision. However, we believe BRCM does not sacrifice simplicity as it is pretty straightforward to implement in practice (though more complex to analyze in theory): the platform first estimates the user attention decaying factors $\{r_i\}$ for each user and then in each user’s interaction session, use them to calculate the reward for each creator sequentially with complexity $O(K)$.
>
> **Q4: Justification of welfare modeling** In this work, we focused on welfare maximization by disentangling this question from other considerations, such as the platform’s revenue, budgets, and attractiveness to creators compared to other outside options. Such modeling methodology is commonly adopted in mechanism design, often for the clarity of analysis. For instance, in the well-known second-price auction, asking the winner to pay any $\alpha(\leq 1)$ fraction of the second highest bid is also a truthful welfare-maximizing mechanism but will change the seller’s revenue. Similarly, in our model, any rescaling of the reward $M$ (or, more specifically, a rescale of the set of functions $f$) does not change the creators’ equilibrium behavior either. Consequently, if the platform wants to upper bound the total reward, that can be easily done by rescaling the rewards. An alternative way to upper bound the total reward in practice could be to incorporate the total budget constraint in the optimization problem in Section 5.2. The platform may consider adding a soft constraint as a regularization term in the objective function. We will include such discussions in our revision. However, we acknowledge that theoretically achieving both total payment minimization and welfare maximization is challenging, and we believe it is more appropriate to defer it to future investigation.
>
> **Q5: Baselines in experiments** In the experiment, the hyperparameter $\beta=0.05$ for $M^3$ baselines is already the optimized choice. We will clarify this in our revision.
>
> **Q6: Why optimized BRCM is better than the optimal one** We provided an explanation in Line 361-366. Simply put, when the mechanism is fixed, the stochastic nature of creators’ responses might slow down the convergence to the welfare-maximizing PNE. However, during the optimization process, the mechanism undergoes dynamic changes with some randomness, leading to an exploration effect that could potentially improve the outcome.

---

> > ### Comment · Reviewer_88Nz · 2023-08-18
> >
> > I thank the authors for addressing my concerns. As noted, I believe that this work extends prior works in this research strand in a non-trivial manner. In my opinion, the paper should have further discussions on the points I've raised, assisting researchers unfamiliar with this research strand to understand the modeling assumptions, weaknesses, future challenges, etc. Based on the authors' rebuttal, I believe they are inclined to do so.

---

> > > ### Author Response · Authors · 2023-08-19
> > > **Response**
> > >
> > > Thank you for taking the time to respond to our rebuttal. We are pleased to hear that you acknowledge the significance of our contribution in extending the existing research strand, and your feedback holds great value in guiding us towards refining our work and making it more comprehensive. We will carefully revise our work to broaden its impact among researchers who may not be acquainted with this specific research domain.

---

### Official Review · Reviewer_TzPr · 2023-07-07

**Soundness:** 3 good
**Presentation:** 3 good
**Contribution:** 2 fair
**Rating:** 6
**Confidence:** 2

**Summary:**

This paper studies the incentives in recommendation systems. Specifically, it studies how to design the platform's reward mechanism to steer the creators' competition towards a desirable welfare outcome. Firstly, it shows a class of mechanisms called "Merit-based Monotone Mechanisms" lead to a constant fraction loss of the welfare. To overcome this loss, it introduces Backward Rewarding Mechanisms (BRMs) and shows that the competition games resulting from BRM induce the strategic creators’ behavior dynamics to optimize any given welfare metric.

**Strengths:**

The paper studies an interesting question in recommendation system. It shows an interesting bad effect of a wide class of mechanisms towards social welfare and then designs another mechanism to overcome the negative result. The theoretical results within the scope of the paper are complete. And there are also empirical experiments.

**Weaknesses:**

The applied value of the model in the paper is lack of justification.

**Questions:**

Can you explain the applied value of the model in this paper?

**Limitations:**

The applied value of the model in the paper is lack of justification.

---

> ### Author Rebuttal · Authors · 2023-08-09
>
> **The applied value of our model** Nowadays more and more content recommendation platforms realize that designing proper incentives for creators is crucial for optimizing social welfare and maximizing their total revenue (such as YouTube and Facebook). However, most of these platforms simply employ rule-based heuristic rewarding mechanisms which are not well understood about their induced content creation dynamics.
>
> Our model formulates the welfare optimization as a mechanism design problem, so that:
> - The long-term effect of different reward mechanisms can be well understood both theoretically and in a simulated environment, thus avoiding a long and expensive feedback circle in online experiments;
> - Our theoretical findings narrow the optimization space by revealing a fundamental limitation of a large class of popular rewarding mechanisms (i.e., $M^3$) and propose an optimization method to search for the optimal mechanism.
>
> Our proposed method has the potential to be directly applied to real-world applications due to the following merits:
>
> - It is easy to implement and straightforward to optimize toward enhancing social welfare (and therefore the revenue of the platforms).
> - It is compatible with any probabilistic recommendation strategy and a variety of personalized welfare metrics.
> - It has good interpretability so that the platform can easily educate creators to digest and follow the designed incentives.

---

> > ### Comment · Reviewer_TzPr · 2023-08-16
> >
> > Thanks authors for the clarification. I keep the original rating.

---

> > > ### Author Response · Authors · 2023-08-20
> > > **Response**
> > >
> > > Thank you for the response. We hope that our clarifications have satisfactorily addressed the reviewer’s questions. Any further feedback or insights that could increase the reviewer’s evaluation of our work are warmly welcomed. Such insights would be invaluable in improving both the quality and impact of our study. Thank you!

---

### Official Review · Reviewer_hvxp · 2023-07-18

**Soundness:** 3 good
**Presentation:** 3 good
**Contribution:** 3 good
**Rating:** 6
**Confidence:** 3

**Summary:**

This paper considers the game played by content creators in recommendation systems, which they call the content creator competition game. This game is centrally defined by a rewarding function M, decided by the platform, which rewards content creators based on how users engage with their content. The paper focuses on how to design this rewarding function in order to maximize user welfare.

They show first that a practically-motivated class of rewarding functions, “Merit-based monotone mechanisms” (M3), lead to losses in user welfare by producing an equilibrium that caters to majority-group users, and fail to cater to minority-group users. Notably, the “necessary welfare loss” is only slightly suboptimal: such mechanisms can still capture a $K/(K+1)$ factor of the optimum, where $K$ is the parameter defining the top-K recommendation policy (though they do make the point that when $K$ is effectively 1 for users who care only about the top recommendation, this ratio can be 1/2). Also notably, they prove this result in a sub-class of creator competition games with the structure of a majority group and several minority groups, all with orthogonal interests (called TvN games - “trend versus niche”).

Next, they introduce a class of rewarding functions called “backward rewarding mechanisms”, which keeps the merit-based property of M3 but drops the monotonicity assumption, trading it for a a set of functions f1,…fn specified by the platform that can be tuned to encourage diversity by making it costly for too many creators to be producing the same kind of content. They show that for TvN games, there exists a backward rewarding mechanism that admits the optimal welfare. They run some simulated experiments with user preferences that constitute a TvN game.

**Strengths:**

- The paper aims to be general: they take almost an axiomatic approach and study an entire class of mechanisms (M3) defined by two main assumptions, which encompass multiple practical mechanisms. They also study an entire class of instances (TvN games).

- They show clean equilibrium results for both classes of mechanisms

- The new class of mechanisms they propose is conceptually interesting — it makes clear why the monotonicity assumption causes problems, and offers a tunable class of algorithms that can help improve user welfare, at least in theory (it remains to be seen whether there is adequate information available in practice to set the parameters of these mechanisms well).

- The paper takes care to make abstract concepts understandable, giving examples and intuition to supplement the math

**Weaknesses:**

1. I do not understand the “monotonicity” property conceptually (described on Lines 43-44 as “the sum of creators’ utilities increase whenever any creator increases her content relevance”). I may be misunderstanding something here, but I interpret this to mean “when one creator benefits, all creators benefit on average”. This doesn’t seem to necessarily reflect an environment in which content creators are competing: Under competition, it could be the case that when a given creator improves her content, it greatly increases her own utility but decreases all other creators’ utilities more in total?

2. Although the paper makes some attempts to justify why the K/(K+1) loss of M3 mechanisms is bad, this doesn’t seem that bad to me, especially given that it emerges from a purely theoretical model in which many abstractions have been made. I felt that the paper oversold the magnitude of this loss in multiple places. It’s also not clear to me that the proposed fixed (BRM) doesn’t have similar loss in other natural sub-classes of games outside TvN games (as their positive results applies only to TvN games). To me, the combination of these two points weakened the motivation for the results sections later in the paper.

Small points about clarity:
- Line 34: what is a “reward signal”?
- Line 41: “and frame a class of prevailing rewarding mechanisms… Merit-based Monotone Mechanism”. It’s not clear what “prevailing” means here.
- In the explanation Lines 50-53, I’m missing a logical step: is “relevance quality” evaluated in terms of the *number* of users who find it relevant? Otherwise, I don’t see how this monotonicity property could cause concentration of creators around majority users’ preferences.

**Questions:**

Questions are encompassed in "weaknesses" section.

**Limitations:**

yes

---

> ### Author Rebuttal · Authors · 2023-08-09
>
> **Clarification of monotonicity** The reviewer’s understanding of monotonicity is correct. In many competing environments, a unilateral improvement of one player’s utility could decrease other players’ utilities but does not necessarily decrease the total utility across all players. For example, when all creators compete for a fixed user attention pool, the total utility of creators remains constant or slightly increased (if user attention increases with more content). And monotonicity summarizes this property of such competing environments. The main message of our negative result is that such competing environments are ineffective for social welfare maximization in online content recommendation platforms because they fail to penalize oversaturation in any particular group of users.
>
> **The significance of negative result** First, we need to clarify that $1/K$ fraction of loss is considered significant in many practical applications, especially commercial platforms, as the span of user attention $K$ is usually not large in many leading content recommendation platforms (e.g. $K\leq 10$). Given that the volume of traffic is large (e.g.,  billions of video impressions per day in TikTok, https://www.usesignhouse.com/blog/tiktok-stats), any small fraction of welfare increase has the potential to create a considerable amount of revenue for the platform, and our proposed mechanisms pave the way for optimizing this marginal gain. Second, our experiment result suggests a constant fraction of welfare loss holds beyond the class of TvN games: Figure 1 (a, c) indicates that when the user population is well separated (e.g., generated from distinct Gaussian clusters in our simulation), the welfare gap induced by BRCM and $M^3$ can be approximately 20% (55+ v.s. 65+ in (a) and 50+ v.s. 60+ in (b)) when $K=5$. Finally, we acknowledge that theoretically generalizing our negative result is an important yet challenging direction, and we consider it as an intriguing future work.
>
> **Clarification of other concepts**  “reward signal” refers to the monetary reward set by the platform. By saying “...prevailing mechanisms” we actually mean “...existing popular mechanisms…” The “relevance quality” means the average relevance matching quality/score over this user population but not the number of users who find it relevant. We will carefully address these notions in the revision of this paper.

---

> > ### Comment · Reviewer_hvxp · 2023-08-12
> > **Response**
> >
> > I have read the authors' response, and I thank them for the clarifications. I have no further questions, and I'm satisfied that the weaknesses I raised are not a significant issue, so I'm inclined to keep my leaning-positive score.

---

> > > ### Author Response · Authors · 2023-08-15
> > > **Response**
> > >
> > > We are glad to hear that our explanations addressed the reviewer’s concerns. We are also excited about the potential of our work in enhancing the efficiency of the online content creation and recommendation ecosystem. Hence, we would be delighted to hear any additional guidance that could increase the reviewer’s evaluation of our work. These would help to significantly enhance the quality and visibility of our research. Thank you!

---

### Author Rebuttal · Authors · 2023-08-09

We thank all the reviewers for the overall positive and informative feedback. In the following, we respond to the questions one by one.

---

### Decision · Program_Chairs · 2023-09-21

**Decision:**

Accept (poster)

**Comment:**

The reviewing team was in agreement regarding the positive assessment of the work. The paper's strong traits are:
1) the model and the results are general for classes of mechanisms, and
2) the problem in and of itself is interesting and extends prior work nontrivially and in meaningful directions.